# Mapping Priority Areas for Connectivity of Yellow-Winged Darter (*Sympetrum flaveolum*, Linnaeus 1758) under Climate Change

Víctor Rincón [1,*] , Javier Velázquez [2] , Derya Gülçin [3] , Aida López-Sánchez [2], Carlos Jiménez [2],
Ali Uğur Özcan [4] , Juan Carlos López-Almansa [2] , Tomás Santamaría [2], Daniel Sánchez-Mata [1]
and Kerim Çiçek [5,6]

1   Faculty of Pharmacy, Department of Pharmacology, Complutense University of Madrid,
    Plaza de Ramón y Cajal, s/n, 28040 Madrid, Spain
2   Department of Environment and Agroforestry, Faculty of Sciences and Arts, Catholic University of Ávila,
    05005 Ávila, Spain
3   Department of Landscape Architecture, Faculty of Agriculture, Aydın Adnan Menderes University,
    Aydın 09100, Turkey
4   Department of Landscape Architecture, Faculty of Forestry, Çankırı Karatekin University,
    Çankırı 18200, Turkey
5   Faculty of Science, Department of Biology, Section of Zoology, Ege University, Izmir 35100, Turkey
6   Natural History Application and Research Centre, Ege University, Izmir 35100, Turkey
*   Correspondence: virincon@ucm.es

**Abstract:** The yellow-winged darter (*Sympetrum flaveolum* Linnaeus, 1758, Odonata), which is associated with high mountain areas, can be considered a flagship species. Due to climate change, its natural range will be negatively affected. In this study, we propose global potential distributions for this species up to the year 2100, considering four time periods (2021–2040, 2041–2060, 2061–2080, and 2081–2100) and three shared socioeconomic pathways (optimistic—SSP245, middle of the road—SSP370, and worst—SSP585), by using an ecological niche model to produce two sets of distribution models (80% to 100% and 60% to 100%). It is foreseen that in the worst of the considered climate scenario (SSP585– 2100 year), the distribution of this species could be reduced by almost half, which could pose a risk for the species and provoke the shift from vulnerable to endangered. An analysis of connectivity has also been carried out for all the studied scenarios by applying the MSPA and PC indices, showing that the core habitat of this species will become more important, which is consistent with the decrease in the distribution range. Over time, the importance of the most valuable connectors will increase, implying a greater risk of some populations becoming isolated.

**Keywords:** ecological connectivity; climate change; MSPA; ecological niche modeling; PC index; dragonfly; Odonata

## 1. Introduction

Climate change is a major concern of the scientific community [1–3]. The consequences of this global impact are manifold; among them, it is likely to have a significant impact on different levels of biodiversity [4–9] and, particularly, on freshwater biodiversity [10]. In the last decade of the 20th century and the first two decades of the 21st century, numerous and diverse studies have been carried out to predict climate change's effects on ecological niches and biodiversity. Thus, changes in the phenology and life cycles of numerous species have been documented [11,12], as well as alterations in their range that may consist of expansions, reductions, or migratory shifts [5,13,14]. In the most extreme situations, climate change can even lead to the extinction of the species [15].

The order Odonata (dragonflies and damselflies) is a group of insects with ideal characteristics and life cycles to be used as bioindicators of ecosystem quality and, thus,

in climate change research [16,17]. They are distributed in freshwater habitats with very specific conditions, and their populations are very sensitive to alterations in environmental conditions such as fluctuations in the water table or flow, air temperature, concentration of pollutants in the water, and water physicochemical characteristics such as electrical conductivity, pH, dissolved oxygen and temperature [18,19].

In this study, the global potential distribution of an odonate species, the yellow-winged darter (*Sympetrum flaveolum* L.), was investigated. It is a dragonfly belonging to the suborder Anisoptera, widely distributed throughout the Palearctic region from Japan to Portugal [20–22]. In southern Europe, and particularly in the Iberian Peninsula, the southwestern limit of its natural range, populations are fragmented and generally associated with mountainous areas. Lowland populations are short-lived and, in most cases, die out after a few years (up to 5–6 years). This can be thought of as an "influx model pattern" followed by decline and disappearance. [20,23–25]. This species also occurs in the southern half of Fennoscandia [26].

The habitat of *S. flaveolum* consists of shallow water areas with abundant vegetation that are usually dry in summer and are neither too eutrophic nor shaded [20,22,25,27,28]. The aforementioned high mountain distribution and its habitat, associated with aquatic environments, make this species particularly susceptible to climate change. In this regard, Warren et al. [29] suggest that a 2 °C increase in average temperature, the maximum limit set at the Paris Summit [30], would make the current areas of distribution unsuitable. Paradoxically, however, a reduction in the depth of high mountain wetlands could improve the status of populations [31]. Therefore, it is foreseeable that the increase in mean temperature brought about by climate change and variations in water tables will lead to significant changes in the distribution of *S. flaveolum*.

Though globally listed as the Least Concern in the IUCN Red List, *S. flaveolum* is considered Vulnerable in some countries, such as Spain [20] or Italy. This fact means that, at least in some peripheral areas of its natural range, it faces a high risk of shifting to Endangered status and, finally, becoming extinct. Likewise, its populations are severely fragmented, and a decrease in the area of distribution and the extent and/or quality of habitat has been observed or inferred [32].

The stenosis of this species, its inclusion in the vulnerable category in the IUCN Red List for some countries, and the need to act on aquatic ecosystems to ensure its conservation justifies the interest in assessing the current status of *S. flaveolum* populations and predicting future scenarios under the pressure of climate change. Likewise, once the evolution of this species is known, it will be possible to infer that of other species with similar ecological values. In that sense, it is urgent to identify effective conservation strategies for protecting the biodiversity of freshwater ecosystems in the climate change scenario in which we are immersed [33].

Thus, in order to predict future scenarios, it is necessary to have a better knowledge of, among other factors, the connectivity between populations, which, according to Bush et al. [34], is a function of the dispersal capacity of the species and the availability of climatic refuge. In this regard, it is worth bearing in mind that *S. flaveolum* is a migratory species [21,35], although it does not present a high dispersal capacity [36].

The present study aims to develop better knowledge of the future situation of *S. flaveolum*, particularly: (i) to predict the potential distribution area of this species in different future scenarios of climate change; and (ii) to study the connectivity within this potential distribution for all scenarios and with two probabilities of appearance (from 60% to 100% and from 80% to 100%) clustering by terrestrial ecoregions with similar connectivity, in order to inform potential conservation measures for this species, which will also contribute to the conservation of other species living in the same habitat.

## 2. Material and Method

### 2.1. Species

The yellow-winged darter [Sympetrum flaveolum (Linnaeus, 1758)] is distributed throughout most of Eurasia from Europe to mid and northern China [37,38]. It occasionally migrates to the United Kingdom [38]. The species is bred in a wide range of stagnant waters. It could be found in peat bogs, waterbodies, garden pools, wetland pools, oxbow lakes, quarry pools, even fishponds, and artificial canals. Adult dragonflies are found from late June to October and peak in August. The nymphs succeed in stagnant water, small, shallow, and rich in vegetation. They are usually found in peat bogs, flooded meadows, and marshy areas, often at higher altitudes [38,39]. The species is a prominent predator and has an important role in the food webs of high-altitude lakes. Therefore, its disappearance would lead to major changes in these food webs [38,39].

### 2.2. Study Area

The study area encompassed the present natural range of *S. flaveolum* and areas where the species could potentially live in the future in Europe and non-tropical Asia. For this study, we hypothesized that all this area is freely accessible to *S. flaveolum* currently and in the future.

### 2.3. Occurrence Data

Future predictions for *S. flaveolum* were made using all available data in the Global Biodiversity Information Facility, from which a total of 19,997 occurrence records were acquired (GBIF 2022: 19,901 records, www.gbif.org (accessed on 1 August 2022), and 96 capture data). These records were verified to be accurate using ArcGIS and georeferenced using the WGS84 coordinate system (v10.7, ESRI, Redlands, CA, USA). We used the tool spThin ver. 0.2.0 [40] to draw a 5-km buffer area around each occurrence record to reduce sampling error that could overestimate the anticipated distribution [39] and to reduce spatial autocorrelation [41,42]. We thinned a total of 19,997 occurrence records to 4837 to represent its presence for each grid cell. We simulated *S. flaveolum's* potential range for both present and future situations [43]. We identified the research area where records of the species exist in order to predict the species' future forecasts. The study region was shielded from climate influences.

#### Climate Data

The WorldClim v2.1 database ([44]; www.worldclim.org), with a geographical resolution of 2.5′ (about 4.7 km), provided the climatic data used in this investigation. The various WorldClim variables were derived from monthly averages of precipitation and temperature for the years 1970 to 2000. The modeling process made use of fifteen recent bioclimatic variables. The removal of four variables (BIO8, BIO9, BIO18, and BIO19) where some spatial artifacts had been found in earlier studies (such as [45,46]) was done.

Using the 'usdm' package [47], we removed the variance inflation factor higher than 5 and used a correlation threshold of 0.75 to lessen the potentially harmful effects that could arise from multicollinearity and high correlation (r>0.75 or −0.75) among the bioclimatic variables [48–52]. The input variables used in this study were BIO2: mean diurnal range (mean of monthly [max temp—min temp]); BIO4: temperature seasonality (standard deviation ×100), BIO5: maximum temperature of warmest month; BIO13 = precipitation of wettest month; BIO14 = precipitation of driest month, and BIO15 = precipitation seasonality (coefficient of variation). For the analysis of the results, two distribution probability ranges (60–100 % and 80–100 %) were considered.

The fit models were projected to five different global circulation models (GCMs): BCC-CSM2-MR [53], CNRM-CM6-1 [54], CNRM-ESM2-1 [55], CanESM5 [56], and MIROC6 [57] to account for an appropriate level of uncertainty in the climate model projections [58]. Future data from the 6th Climate Model Intercomparison Project (CMIP6, www.wcrp-climate.org/wgcm-cmip/wgcm-cmip6) with three shared socioeconomic pathways were acquired

for the periods 2021–2040, 2041–2060, 2061–2080, and 2081–2100. (SSPs) (optimistic—SSP245, middle of the road—SSP370, and worst—SSP585).

### 2.4. Methodology

*Phase 1. Sampling and data collection on the distribution of the target species*

To know the global distribution of *Sympetrum flaveolum*, location points were compiled using the Global Biodiversity Information Facility database. For climatic data, the climate models of worldclim were used.

*Phase 2. Ecological niche modeling*

Using an ensemble method in the sdm package [40] in the R v3.6.3 environment, we created ecological niche models to project the current and future habitat suitability of *S. flaveolum*. The generalized linear model (GLM; [41], boosted regression trees (BRT) [42], random forests (RF) [43], and maximum entropy (MaxEnt) [44] with 10,000 randomly selected pseudo-absences were five algorithms that we implemented using various approaches. According to Naimi and Araújo [41], we used the sdm package's usual parameterization to run all the algorithms. The small sample size necessitated the use of the subsample and bootstrapping resampling methods [45], which were divided into subsets of 70-30% for model calibration and testing. A true skill statistic (TSS) and the area under the receiver operating characteristic (AUC) were calculated for each model, and each model was run ten times. To assess the species' adaptability to its current environment at the time of analysis, we created 50 distinct models (5 algorithms × 1 resampling method × 10 replications). To create ensemble models for each scenario, we chose the models with TSS > 0.7 and AUC > 0.9 as the best. The best models were assembled using the mean of predicted presence-absence values technique, which involves converting the expected probability of occurrences to presence-absence using a threshold before averaging. Following that, the chosen models were projected into current and future circumstances. As a result, we implemented 60 projections (5 GCMs, 3 SSPs, and 4 time periods) and generated ensemble rasters for the GCM scenarios and periods. The outputs indicate habitat suitability on a scale of 0 (unsuitable) to 1 (suitable) (suitable). We converted the ensemble suitability models into binary maps of acceptable environmental conditions and used them by maximizing the sum of sensitivity and specificity (maxSSS), as Liu et al. [46] proposed. The RasterVis package was used to illustrate the results [48].

*Phase 3. Comparison of distribution areas*

In this phase, changes in the distribution area between the current situation and the expected situation for all the proposed scenarios were analyzed.

*Phase 4. Calculation of connectivity using the MSPA and the PC index*

This phase aims to compare connectivity by calculating the morphological spatial pattern analysis (MSPA) [47], which measures structural connectivity through the number of connecting elements, and the probability of connectivity index (PC) [47], which measures the importance of each of the connection elements previously analyzed, for all climate scenarios and with the two probability ranges, resulting in a total of 26 possible situations.

### 2.5. Statistical Analysis

In order to group the terrestrial ecoregions in clusters with similar connectivity (number of yellow-winged darter links), we used principal component analysis (PCA) [59]. First, we developed a PCA analysis with the number of links from terrestrial ecoregions obtained in the previous analysis according to the current situation and climate change scenarios: optimistic—SSP245, middle of the road—SSP370, and worst—SSP585 for 2040, 2060, 2080 and 2100. Then, we classified all terrestrial ecoregions (ordered by PCA) according to their similar connectivity across the different climate change scenarios. The hierarchical classification was performed using Ward's criterion on the selected principal components [60].

R. 4.2.1 (R Development Core Team 2022) with the packages "FactoMineR" [61], "factoextra" [62], and "vegan" [63] were used for data processing and statistics.

## 3. Results

Overall, our ecological niche models (ENMs) have an average AUC of 0.936 (SD = 0.036) and an average TSS of 0.775 (SD = 0.115). Our ENM for present-day conditions indicates that the habitats suitable for *S. flaveolum* spread across most of Europe, western Siberia, northern Anatolia, the Caucasus, the Himalayas, northern Japan, Sakhalin, and Kamchatka Peninsula (Figure 1). According to the occurrence record, it is also quite frequent in other areas where our model indicates a low probability of occurrence, such as Korea or Mongolia.

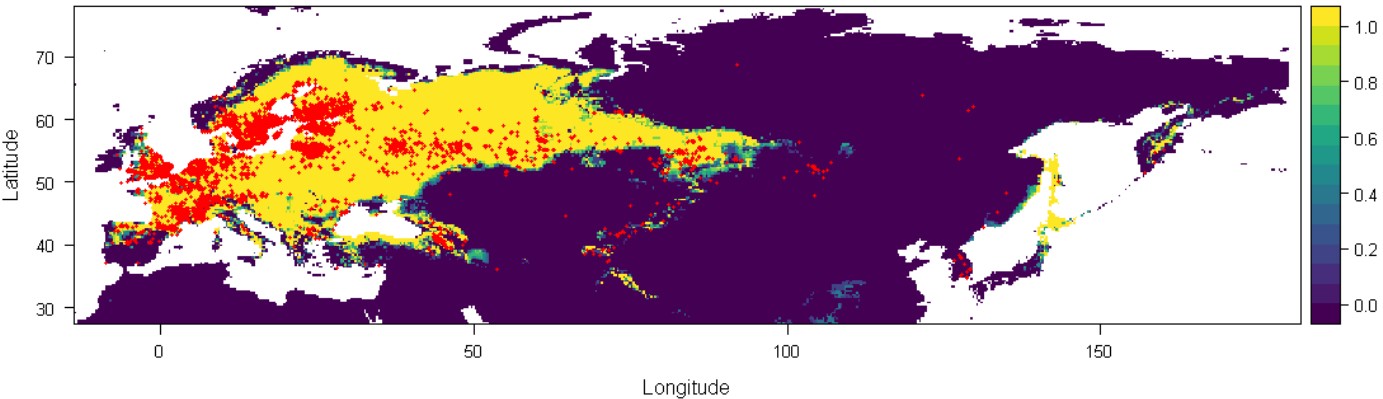

**Figure 1.** Average prediction of climate habitat suitability maps for *Sympetrum flaveolum* projected to the present day. Red dots show occurrence records. The probability of occurrence ranges from 0 (dark purple, low probability) to 1 (yellow, highest probability).

Our results show that the habitat suitability of the species is explained by temperature seasonality (BIO4, 29%), precipitation of the driest month (BIO14, 22%), precipitation seasonality (BIO15, 19%), the maximum temperature of the warmest month (BIO5, 19%), mean diurnal range (BIO2, 9%), and precipitation of wettest month (BIO13, 8%). They also predict that the suitable habitats for this species will shift towards the north and will disappear up to 2100, under future climate scenarios, in southern Europe, Anatolia, the Caucasus, the southernmost area of western Siberia, and Japan, thus comprising all the southern limit of its present range (Figure 2). At the same time, some northern areas that are currently unsuitable for *S. flaveolum* will become suitable, as would be the case in northwestern Siberia and Chukotka.

Tables 1 and 2 show the changes in the potential distribution of *Sympetrum flaveolum* for the different study scenarios. For both distributions (80–100% and 60–100%), the potential area will decrease until it reaches 60.93% and 63.4%, respectively, for the year 2100 in the worst possible scenario. With scenario SSP245, affection would be initially much lower since its distribution would be reduced, in the year 2080, by 3% (80–100%) and just 0.7% (60–100%), though by the year 2100, affection will increase, with distribution been reduced to 91.81% (80–100%) and 93.88% (60–100%). In an intermediate position is scenario SSP370, the only scenario in which the area would decrease in all years and in which the area in the year 2100 would decrease by a quarter for both distributions.

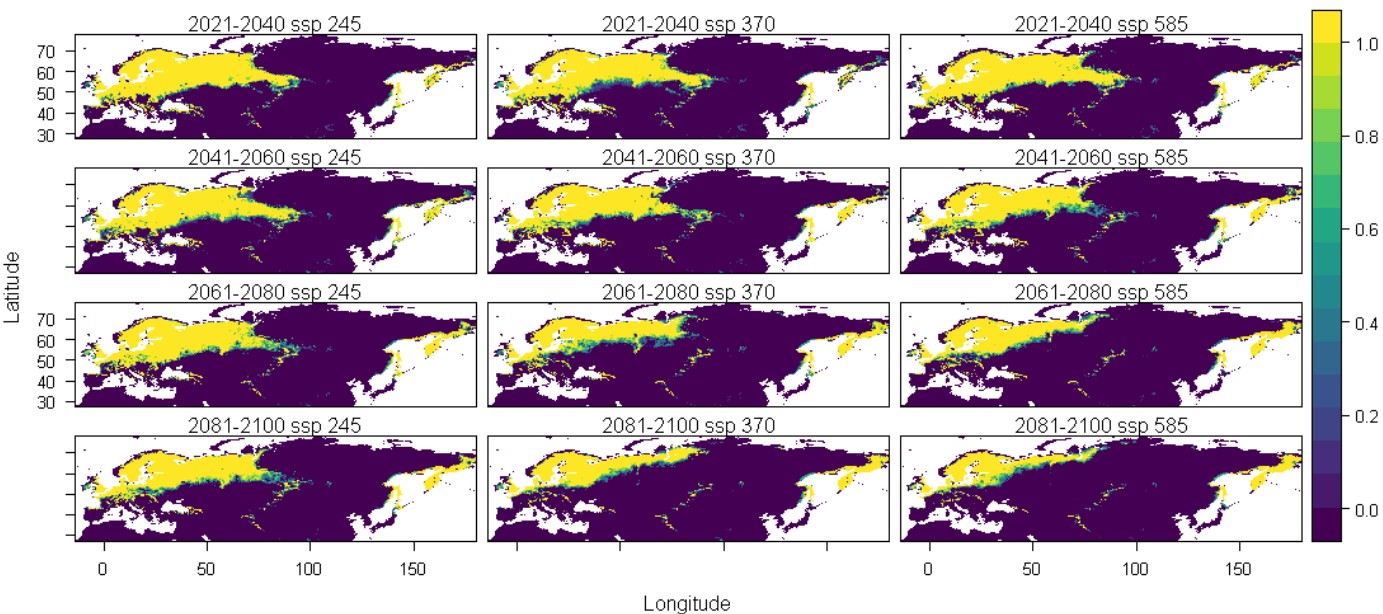

**Figure 2.** Average prediction of climate habitat suitability maps for *Sympetrum flaveolum* under future climate scenarios. Average projections are presented for each of four time periods (2021–2040, 2041–2060, 2061–2080, and 2081–2100) and three shared socioeconomic pathways (optimistic—SSP245, middle of the road—SSP370 and worst—SSP585). The probability of occurrence ranges from 0 (dark purple, low probability) to 1 (yellow, highest probability).

**Table 1.** Evolution of the estimated potential distribution area of *Sympetrum flaveolum* in absolute and relative values (Present—2100). 80–100%.

| Scenario | | Area (km²) | % |
|---|---|---|---|
| Present-time | | 31,105,874 | 100 |
| 2040 | SSP245 | 32,589,949 | 104.77 |
| | SSP370 | 29,751,246 | 95.65 |
| | SSP585 | 31,792,341 | 102.21 |
| 2060 | SSP245 | 30,437,912 | 97.85 |
| | SSP370 | 27,751,750 | 89.22 |
| | SSP585 | 27,491,675 | 88.38 |
| 2080 | SSP245 | 30,411,851 | 97.77 |
| | SSP370 | 24,836,400 | 79.84 |
| | SSP585 | 23,258,831 | 74.77 |
| 2100 | SSP245 | 28,557,897 | 91.81 |
| | SSP370 | 23,331,819 | 75.01 |
| | SSP585 | 18,952,658 | 60.93 |

**Table 2.** Evolution of the estimated potential distribution area of *Sympetrum flaveolum* in absolute and relative values (Present—2100). 60–100%.

| Scenario | | Area (km²) | % |
|---|---|---|---|
| Present-time | | 32,805,232 | 100.00 |
| 2040 | SSP245 | 34,373,533 | 104.78 |
| | SSP370 | 32,005,322 | 97.56 |
| | SSP585 | 33,772,939 | 102.95 |

**Table 2.** *Cont.*

|  | Scenario | Area (km²) | % |
|---|---|---|---|
| 2060 | SSP245 | 32,649,735 | 99.53 |
|  | SSP370 | 30,047,687 | 91.59 |
|  | SSP585 | 29,845,603 | 90.98 |
| 2080 | SSP245 | 32,575,874 | 99.30 |
|  | SSP370 | 27,262,806 | 83.11 |
|  | SSP585 | 25,114,100 | 76.56 |
| 2100 | SSP245 | 30,797,932 | 93.88 |
|  | SSP370 | 25,166,930 | 76.72 |
|  | SSP585 | 20,797,436 | 63.40 |

### 3.1. MSPA Index

The distribution of the components of the MSPA, according to the results provided by Guidos software, is shown in Figure 3 for the current distribution and in Appendices A.1 and A.2 for the potential distribution under the different considered scenarios. Comparing these situations, it is found that, in all the cases, the area occupied by cores decreases notably, throughout the study period, especially in peripheral zones, except in the Russian Far East. Simultaneously, there is an increase in the number of islets in southern areas currently occupied by this species and a significantly higher number of bridges.

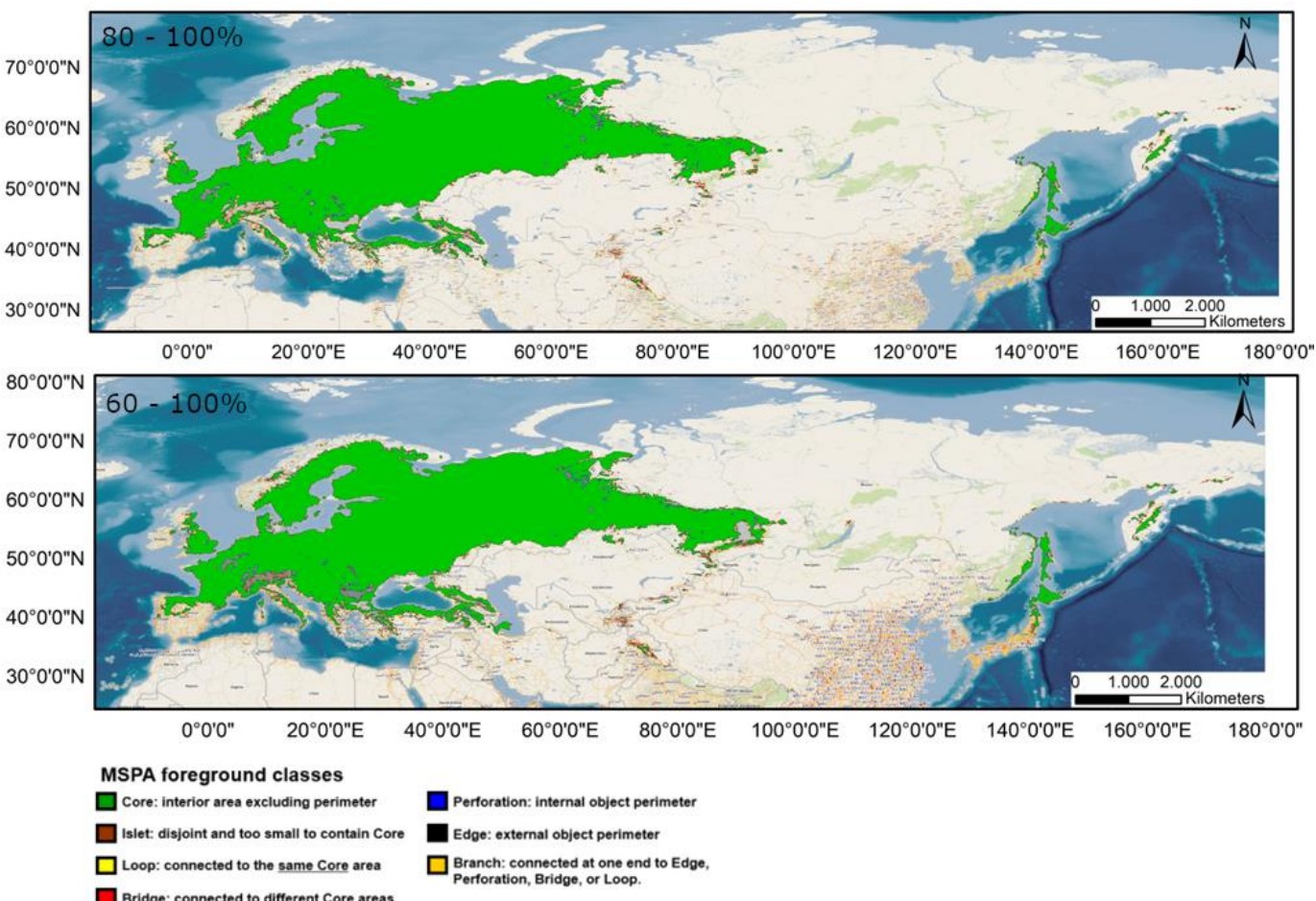

**Figure 3.** Results of the MSPA analysis for the current situation for distributions 80–100% (**up**) and 60–100% (**down**).

It should be noted that all possibilities show an increase in the number of bridges compared to the current distribution.

As these bridges link habitat patches and serve as vital functional dispersal corridors, their gradual loss until 2100 has a detrimental impact on the maintenance of functional connectedness [64,65].

The critical areas for both distributions have also been located and are shown in Appendices A.3 and A.4.

In addition, the PCA revealed an ordination of terrestrial ecoregions according to the different climate change scenarios considered in this study (Figure 4). The two first principal components (dimensions) explained more than 85% of the cumulative variance for both distributions, 60–100% and 80–100% (Figure 4i,ii, respectively). Finally, using the PCA hierarchical classification, we classified the ecoregions into four clusters with similar connectivity across the different scenarios of climate change and by distribution (60–100% Figure 4ii and 60–100% Figure 4iv). Most of the ecoregions were included in clusters 1 and 2 (Figure 4ii,iv; Appendices A.5 and A.6). However, Alps Conifer and mixed forests were exclusively included in cluster 4 for both distributions (Appendices A.5 and A.6), having different connectivity (number of links) that other terrestrial ecoregions by the different scenarios of climate change. Cluster 3 also included a few terrestrial ecoregions: Bering Tundra, Scandinavian Montane Birch Forest and Grasslands, and West Siberian Taiga. The tables with the calculation variables for the different clusters can be found in Appendices A.7 and A.8.

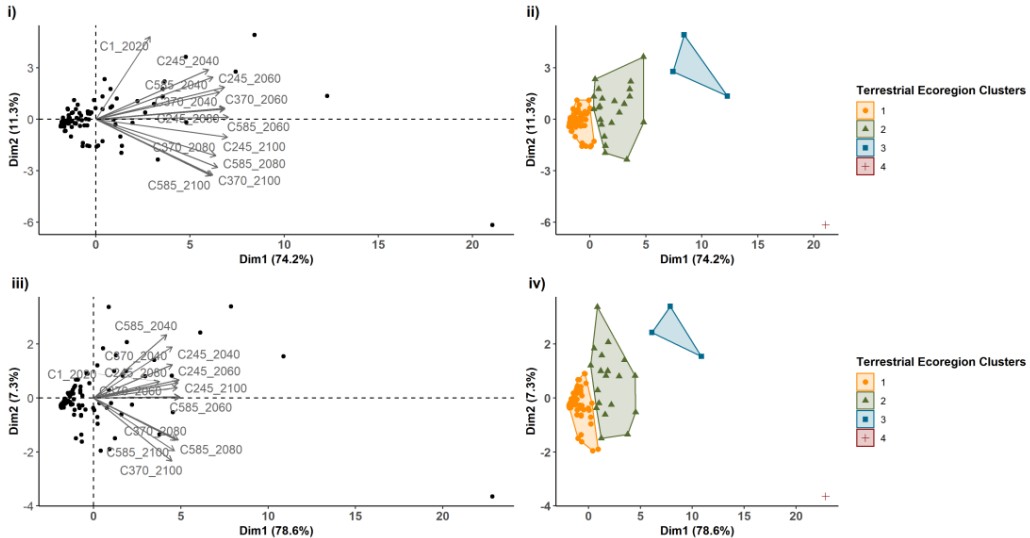

**Figure 4.** Connectivity terrestrial ecoregion ordination and cluster classification. Principal component analysis ((**i**) for 60–100% distribution and (**iii**) for 80–100% distribution) and hierarchical classification ((**ii**) for 60–100% distribution and (**iv**) for 80–100% distribution) focusing on link data numbers from the terrestrial ecoregions according to three shared socioeconomic pathways (optimistic—SSP245, middle of the road—SSP370, and worst—SSP585 in years 2040, 2060, 2080, and 2100. The symbols correspond to terrestrial ecoregions for points and thirteen climate change scenarios for grey arrows (**i**) for 60–100% distribution and (**iii**) for 80–100% distribution) and to terrestrial ecoregions grouped in four clusters (with different colors in the graph) with similar connectivity (**ii**) for 60–100% distribution and (**iv**) for 80–100% distribution). PCA1 and PCA2 explained 74.2% and 11.3% of the variance for 60–100% distribution (**i**), respectively, and PCA1 and PCA2 explained 78.6% and 7.3% of the variance for 80–100% distribution (**iv**), respectively.

*3.2. PC Index*

The study of the dPC index ranks the nodes and links on the map according to how much they contribute to connectivity [66]. As can be observed in Appendices A.1 and A.2, since core regions are usually so huge that we cannot use them to support management

methods for boosting connectivity, we are focusing on corridors, which are also the most important structures in terms of connectivity [64–67].

Tables 3 and 4 show the variation in the importance of connectors in the different considered scenarios and throughout the study period. According to these data, in the 80–100% distribution, the number of connectors increases in the first decades of the study period, reaching a maximum for SSP535 by the year 2040 (7370 connectors) and for SSP245 (6048 connectors) and SSP370 (6873 connectors) by the year 2060. Later on, the number of connectors decreases, reaching the minimum by the year 2080 for SSP245 (5808 connectors) and by the year 2100 for SSP370 (4281 connectors) and SSP535 (3499 connectors). In the 60–100% distribution, a similar pattern was present, with a maximum in the number of connectors by the year 2040 (SSP370 and SSP535) and 2060 (SSP245) and a minimum in 2100 for the three scenarios.

**Table 3.** Variation in the importance of connectors with the 80–100% distribution.

| Scenario | | PC Sum Links | Max | Number | Median |
|---|---|---|---|---|---|
| Present-time | | 54.97824 | 0.588259 | 5136 | 0.005591076 |
| 2040 | SSP245 | 7.476563 | 0.073731 | 5919 | 0.002369475 |
| | SSP370 | 16.674742 | 0.161898 | 5732 | 0.004110193 |
| | SSP585 | 4.194349 | 0.231289 | 7370 | 0.000711398 |
| 2060 | SSP245 | 115.543238 | 0.843213 | 6048 | 0.015330767 |
| | SSP370 | 13.302049 | 0.0956 | 6873 | 0.02986639 |
| | SSP585 | 178.388911 | 0.497718 | 6464 | 0.002514614 |
| 2080 | SSP245 | 63.954432 | 0.834766 | 5808 | 0.008266973 |
| | SSP370 | 6.21363 | 0.268315 | 5414 | 0.011441741 |
| | SSP585 | 27.462305 | 2.307007 | 5004 | 0.002787564 |
| 2100 | SSP245 | 195.926497 | 0.509482 | 5907 | 0.005591104 |
| | SSP370 | 1.52086 | 0.038689 | 4281 | 0.133939254 |
| | SSP585 | 142.119542 | 10.481495 | 3499 | 0.001183118 |

**Table 4.** Variation in the importance of connectors with the 60–100% distribution.

| Scenario | | PC Sum links | Max | Number | Median |
|---|---|---|---|---|---|
| Present-time | | 29.733343 | 0.678515 | 5318 | 0.01070449 |
| 2040 | SSP245 | 15.873111 | 0.168677 | 6699 | 0.00126315 |
| | SSP370 | 28.894654 | 0.168677 | 7030 | 0.00290906 |
| | SSP585 | 4.968406 | 0.162946 | 6984 | 0.00056911 |
| 2060 | SSP245 | 105.153733 | 1.078804 | 6859 | 0.01910437 |
| | SSP370 | 209.034867 | 0.72389 | 6999 | 0.00193541 |
| | SSP585 | 17.448906 | 0.158492 | 6939 | 0.02759729 |
| 2080 | SSP245 | 53.900665 | 0.758437 | 6520 | 0.01101144 |
| | SSP370 | 71.11042 | 0.564638 | 6215 | 0.0011477 |
| | SSP585 | 13.438845 | 3.247807 | 4821 | 0.00548807 |
| 2100 | SSP245 | 33.691993 | 0.244705 | 6026 | 0.03316853 |
| | SSP370 | 633.264791 | 10.453527 | 4728 | 0.00035526 |
| | SSP585 | 4.544358 | 0.143843 | 3841 | 0.04061719 |

Usually, the maximum value of ecological corridors varied between 0.073731 (SSP245, the year 2040) and 2.307007 (SSP585, the year 2080) with the 80–100% distribution and between 0.143843 (SSP585, the year 2100) and 3.247807 (SSP585, the year 2080) with the 60–100% distribution. In the year 2100, however, a value of 10.48 is found for the maximum value of ecological corridors for the 80%–100% distribution for scenario SSP585, and a very similar value is found for the same year 2100 in the 60%–100% distribution for scenario SSP370 (10.45), suggesting that in 2100, with the reduction in surface area and

potential isolation of the cores, the most crucial connectors assume a greater value to prevent fragmentation.

Analyzing the global set of connectors, in the 80–100% distribution, the value of importance is multiplied by almost four in scenario SSP245 for the year 2100. For the 60–100% distribution, this increase occurs for scenario SSP370, which corresponds to a 2100% increase compared to the importance of the current scenario. Such high increases indicate that the fragmentation that will occur in the different scenarios is maximum, so the importance of the connectors is increased to reduce the possible fragmentation caused by climate change.

In order to determine the PC index in smaller areas, PC values have been calculated for ecoregions in which there is potential distribution. Figure 5 shows, as an example, the evolution of the PC indices for the ecoregions and the different clusters previously obtained.

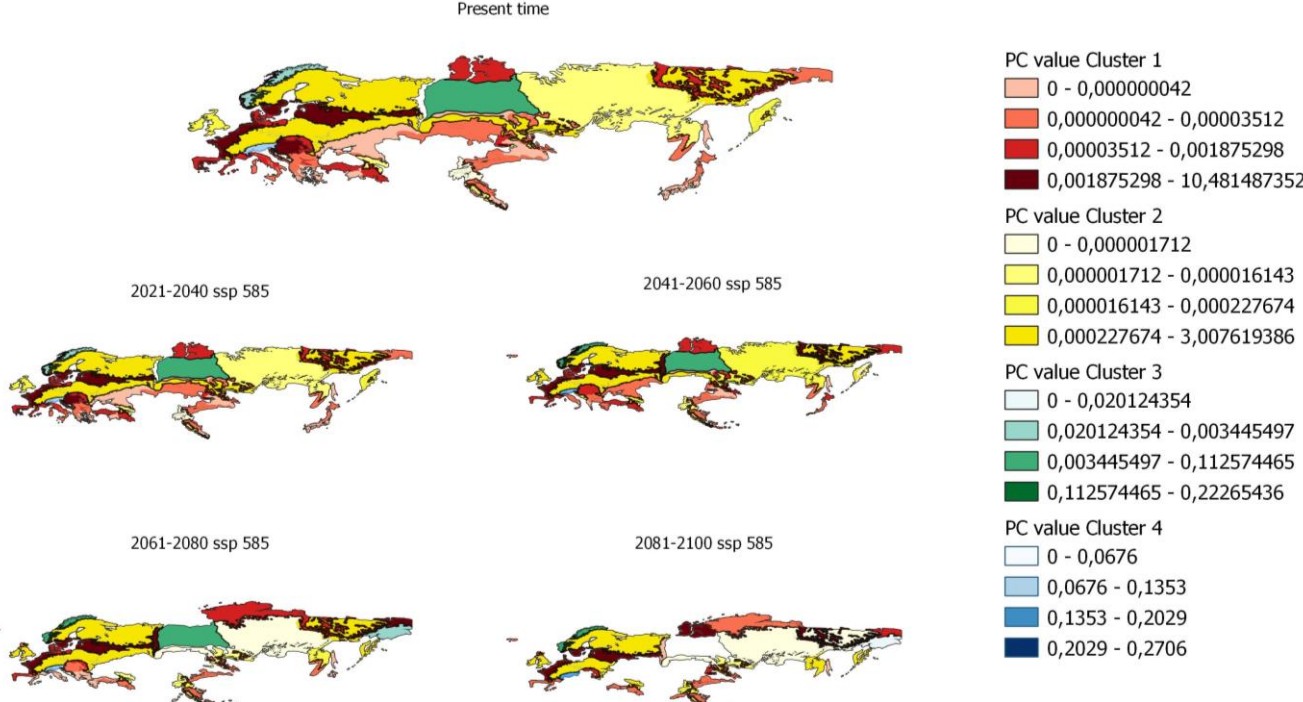

**Figure 5.** Evolution of pc values by ecoregion for scenario 585 in the distribution from 80% to 100%.

Maps of all the index values calculated for the different scenarios and distributions are shown in Appendices A.5 and A.6.

## 4. Discussion

While assessing connections between habitats in a landscape matrix, changes that may occur in land cover over time and how species may spread against bioclimatic variables are often ignored [49–51,68]. However, the responses of species to global climate change have been accepted as the most important environmental factor that determines the main characteristics of habitats and their distribution areas [53,54]. In this context, understanding the direction and magnitude of species responses is important for species conservation and sustainability [55,56]. Since climate change differentiates the bioclimatic demands of species under optimal conditions, it also causes changes in their geographical distribution [57,68–71]. This change is widely linked to increased temperatures and decreased precipitation during the growing season [72]. Every 1 °C change in temperature moves ecological regions around the world about 160 km. Thus, for example, if the climate warms by 4 °C in the next century, species in the northern hemisphere may need to move 500 km north (or 500 m higher) [73]. Many studies that refer to the impact of climate

change on species have investigated the effects of global temperature rise, confirming that species will migrate to the poles (higher latitudes) and higher altitudes as a result [74]. In addition, it has been predicted that the geographic ranges of species will expand, shift or contract [75]. While some studies indicate that certain species may become stronger against climate resistance in the future, it is predicted that some will experience habitat loss, which will negatively affect biodiversity [76–78]. Furthermore, it has been reported that rapid climate change may put pressure on relict species and cause species extinction [78–80]. Climate-related variables such as temperature and precipitation are important for the effects on species survival, distribution, and other characteristics, as well as for the species composition of natural ecosystems and the future of terrestrial ecosystems [81,82].

Our study is consistent with the aforementioned studies, suggesting that the natural range of *S. flaveolum* will reduce significantly in all the proposed scenarios, with this loss being particularly large (up to 40%) in the SSP585 scenario. Furthermore, as has been predicted in other species, the distribution area will also shift northward. Consequently, this species will virtually disappear from the Mediterranean Basin and other southern locations and will spread to northern areas of the Eurasian continent.

The results of the MSPA analysis and PC index showed a loss of connectivity in *S. flaveolum* patches, particularly in its southernmost range. Lack of landscape connectivity can isolate habitat patches that affect gene flow, among other ecological processes. Greater connectivity increases the ability of species to migrate to new regions in the face of climate change and reduces the likelihood of extinction. For this reason, greater connectivity may increase the chances of many organisms surviving under changing climatic conditions. Consequently, this loss of connectivity will negatively affect the populations of the yellow-winged darter and will pose a serious threat to the survival of this species in southern Eurasia.

Due to the strong dispersal capacity of dragonflies in general, changes in the current climate and resource availability primarily affect how they are distributed. This is because dragonflies can track changing climatic and environmental circumstances owing to their flying ability. Olsen et al. [83] stated that dragonflies are often influenced by habitat specialization (species vulnerability to habitat loss and fragmentation [84] or linked dispersal limitation. Previous studies confirm this statement and highlight that extreme habitat specialization can be more effective than dispersal ability, particularly for permanent running water species. The differences between nodes and links in this study can be a reason for either extreme habitat specialization or reduced dispersal ability.

Mountain chains in the European topography can act as barriers for odonate species; therefore, wide river plains can be regarded as corridors. This can be understood from the maps produced in this study. The northern side of the Iberian Peninsula or northern Europe is highly affected by climate change making these areas critical. Geostatistical analysis of the data from the critical detection areas supports this.

We discover that species in permanent water habitats, including both rushing and standing waters, move north to a far lesser extent than those that are adapted to seasonally dry habitats. This suggests that transient waters support the diversity of dragonflies and serve as stepping stones for the spread of generalist species [81]. In comparison to species suited to permanent flowing water environments, species adapted to permanent standing water or transient water habitats, which are less persistent in time and space, spread more effectively [85].

From the point of view of potential distribution, and based on this study, it is more advisable to use the 60% to 100% range since the connectivity shown by this distribution is included within the 80% to 100% range, and the discontinuous zones show where the fragmentation risks really are.

## 5. Conclusions

Experimental studies that use ecological niche modeling predict significant changes in species distributions in response to climate change. As habitat fragmentation can hinder

species range changes, maintaining wildlife corridors may be of increasing importance in enhancing climate resilience for species survival. Therefore, identifying degrees of connectivity between habitats play a vital role in adapting to changing climatic conditions.

In this study, current, potential, and future connectivity changes in *S. flaveolum* were predicted by combining an ecological niche model and an ecological connectivity approach. Besides determining suitable habitats for the species, we identified priority areas for connectivity relevant to the sustainability of *S. flaveolum*. Our approach provides a robust and practical tool to optimize biodiversity conservation objectively. Further study can integrate land use/land cover changes into our method and make a broader interpretation of the species distribution.

**Author Contributions:** Conceptualization, V.R., J.V. and D.G.; methodology, V.R., J.V., D.G. and A.L.-S.; validation D.G., A.U.Ö. and K.Ç.; formal analysis, J.V., K.Ç. and A.L.-S.; investigation, V.R., J.V. and D.G.; data curation, A.L.-S. and K.Ç.; writing—original draft preparation, C.J., J.C.L.-A., D.G. and J.V.; writing—review and editing, V.R., J.V., D.G. and A.L.-S.; visualization, A.U.Ö. and D.G.; supervision, J.V., T.S. and D.S.-M. project administration, J.V. All authors have read and agreed to the published version of the manuscript.

**Funding:** This research received no external funding.

**Data Availability Statement:** The data of the current research is available from the corresponding author on request.

**Conflicts of Interest:** The authors declare no conflict of interest.

## Appendix A

*Appendix A.1. MSPA Analysis under Different Climate Scenarios (60–100%)*

2021-2040 SSP 245

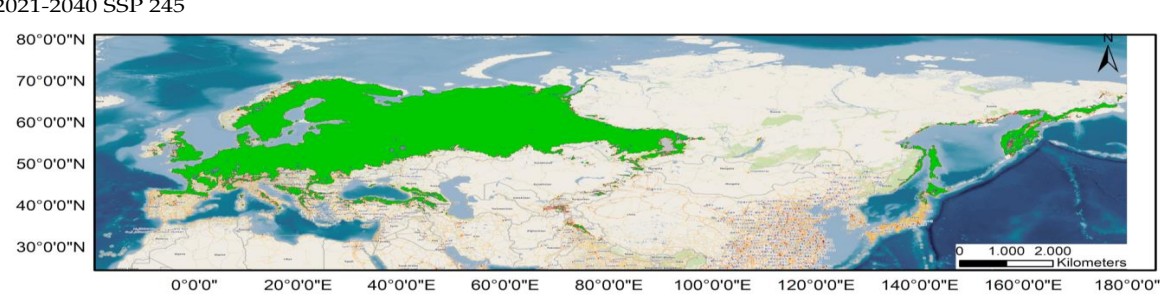

2041-2060 SSP 245

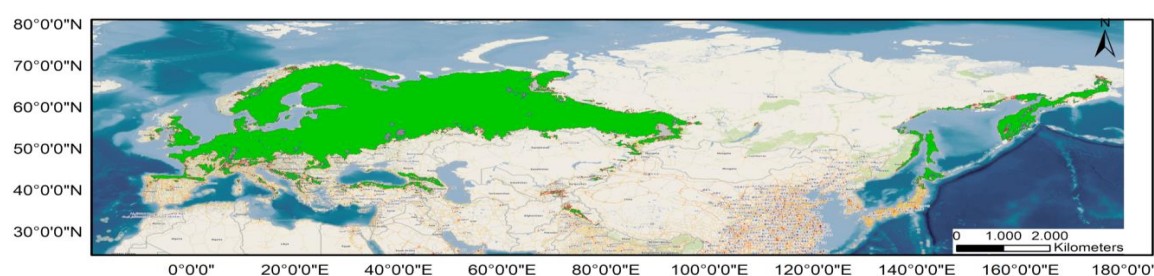

2061-2080 SSP 245

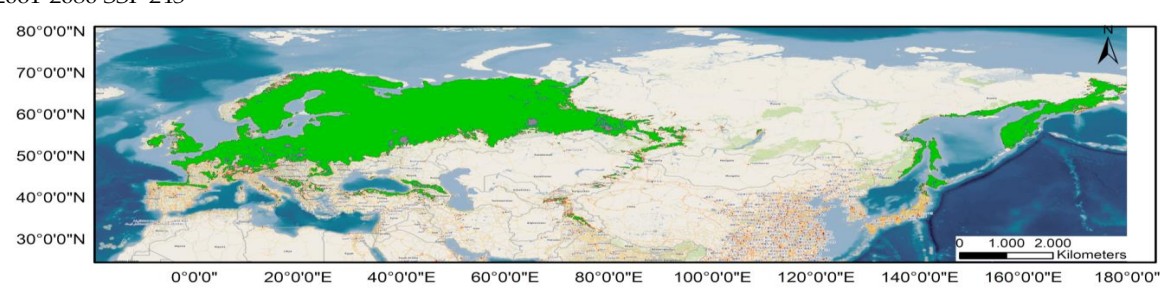

2080-2100 SSP 245

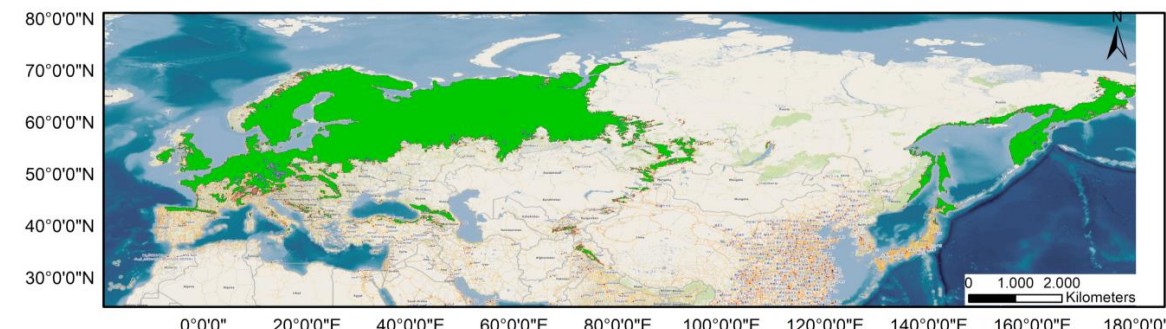

2021-2040 SSP 370

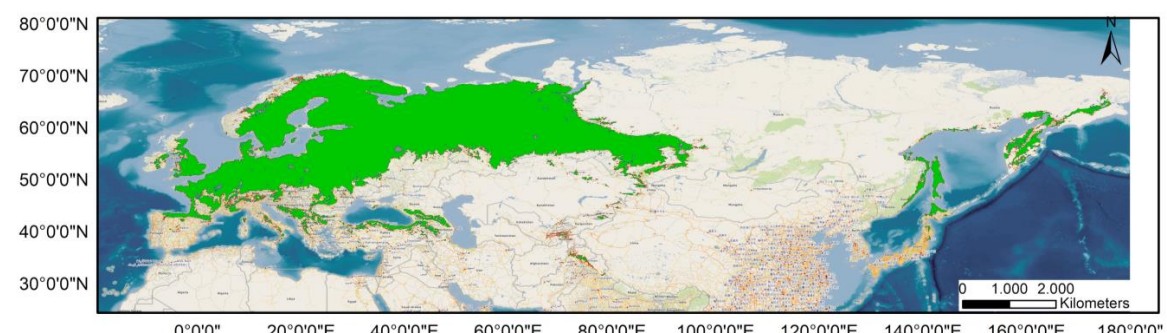

2041-2060 SSP 370

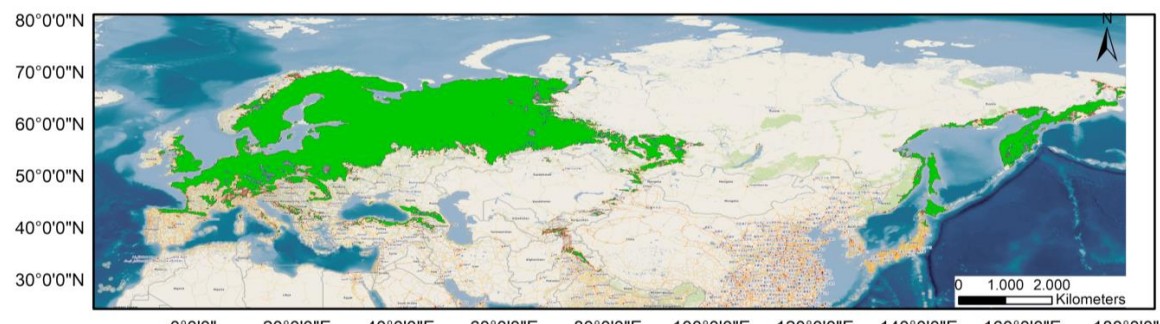

2061-2080 SSP 370

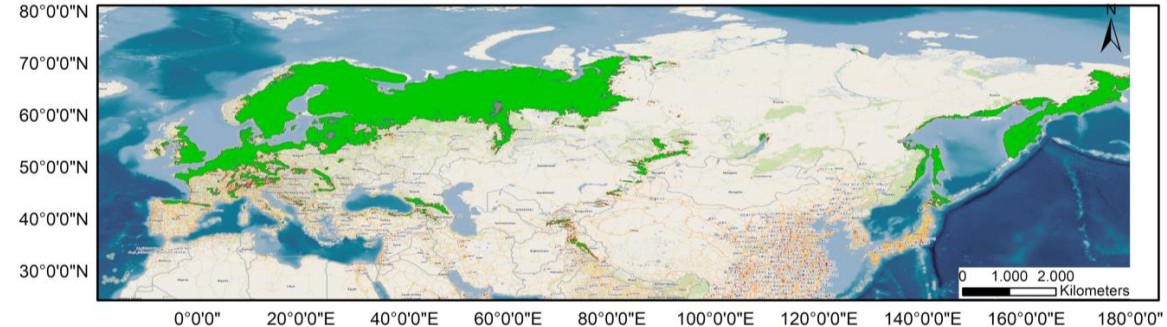

2080-2100 SSP 370

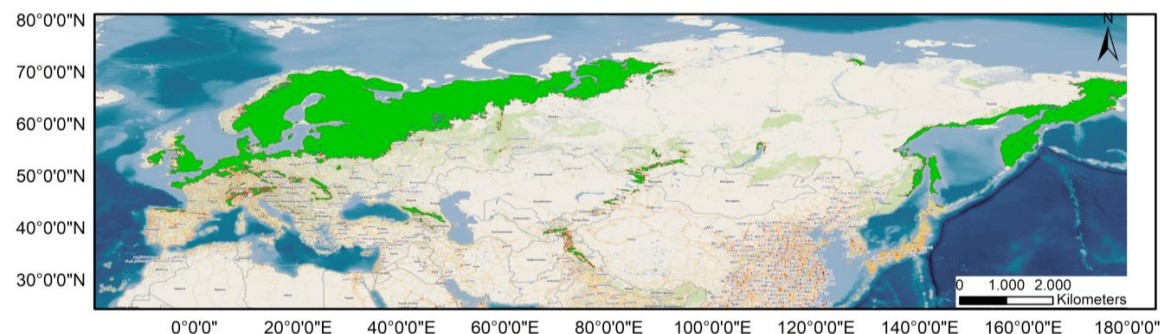

2021-2040 SSP 585

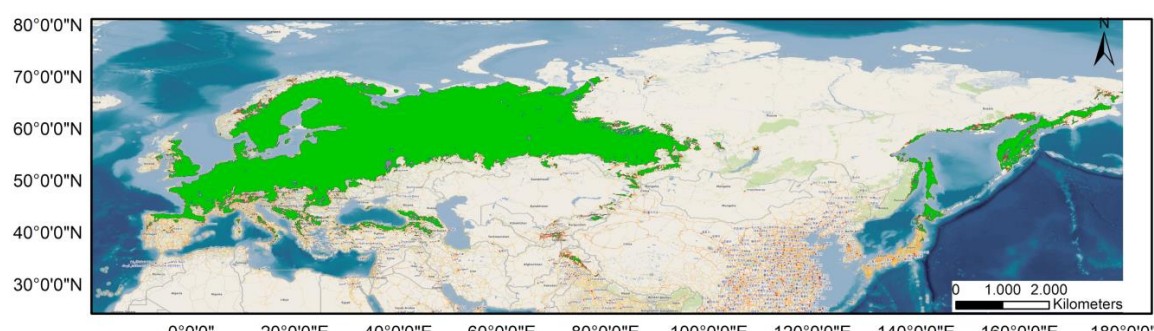

2041-2060 SSP 585

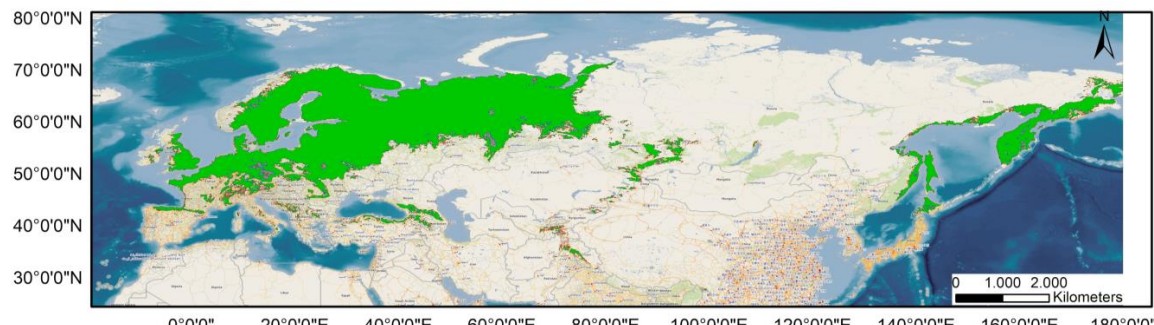

2061-2080 SSP 585

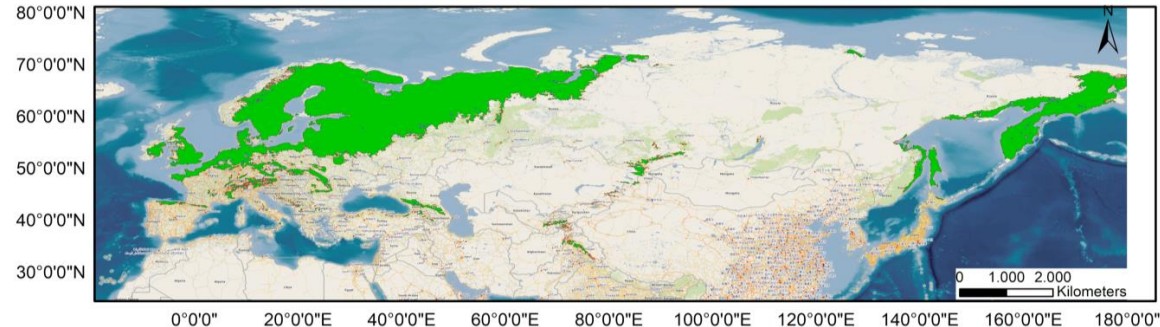

2080-2100 SSP 585

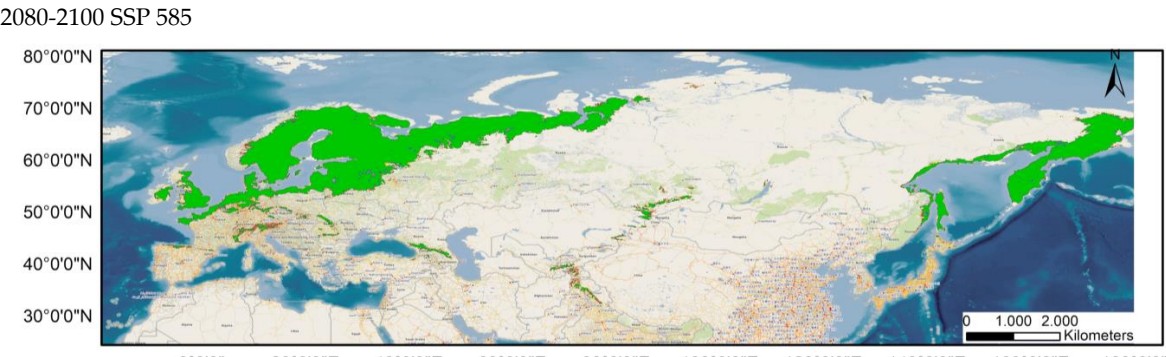

*Appendix A.2. MSPA Analysis under Different Climate Scenarios (80–100%)*

2021-2040 SSP 245

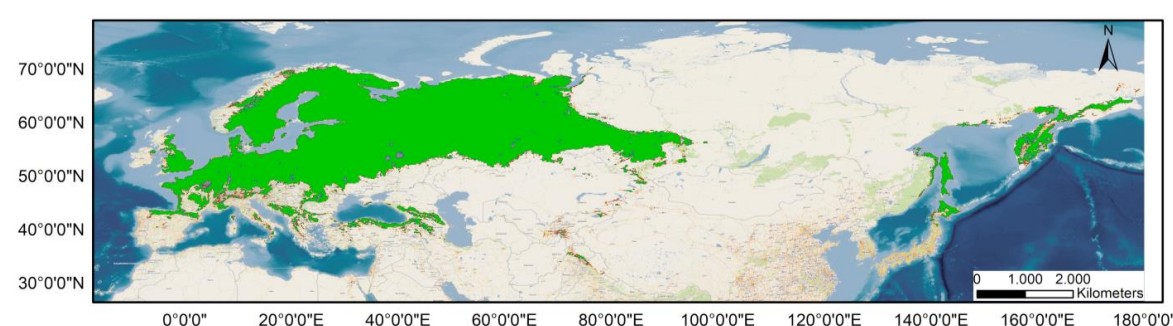

2041-2060 SSP 245

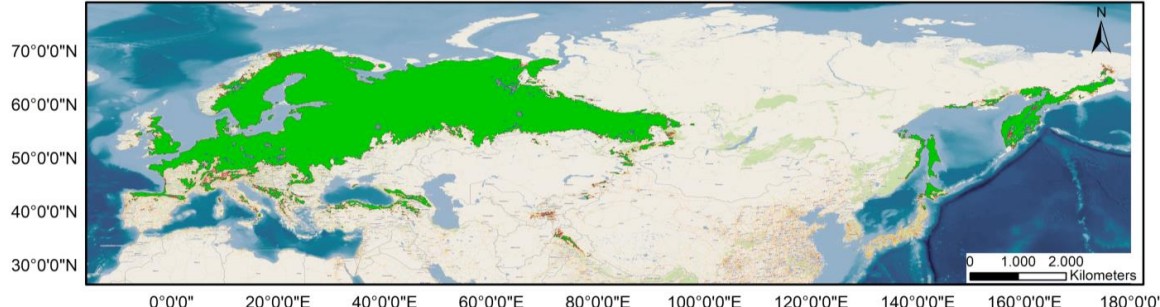

2061-2080 SSP 245

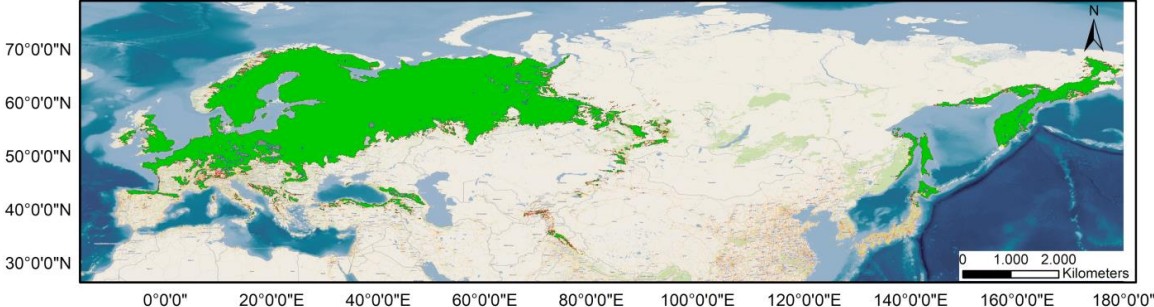

2080-2100 SSP 245

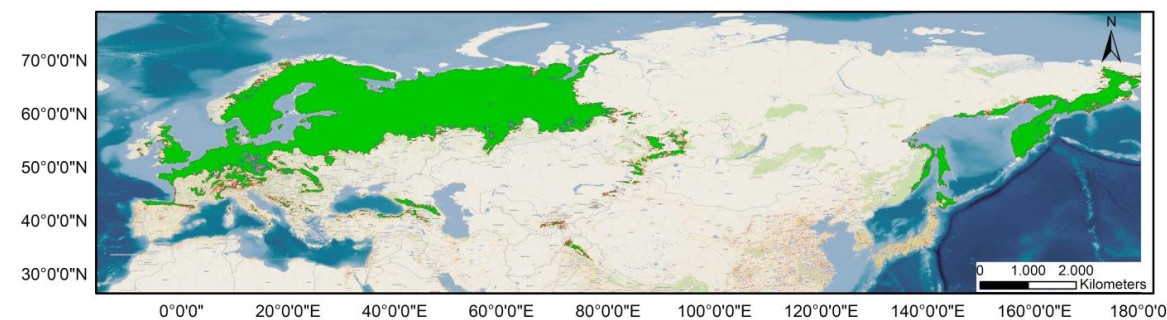

2021-2040 SSP 370

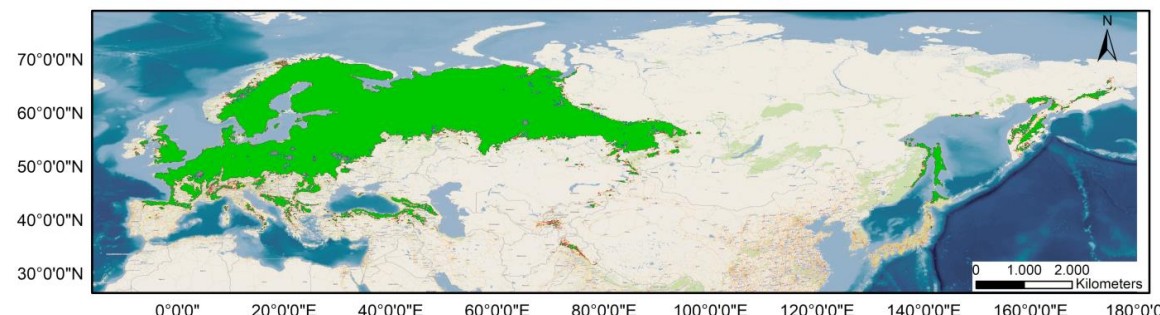

2041-2060 SSP 370

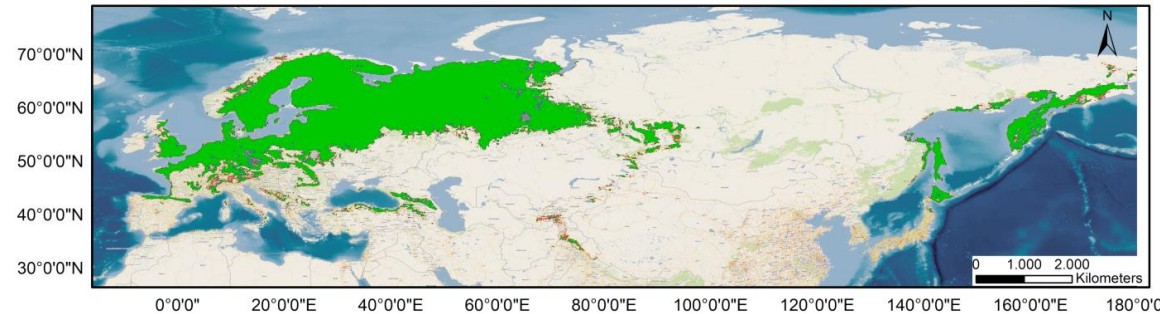

2061-2080 SSP 370

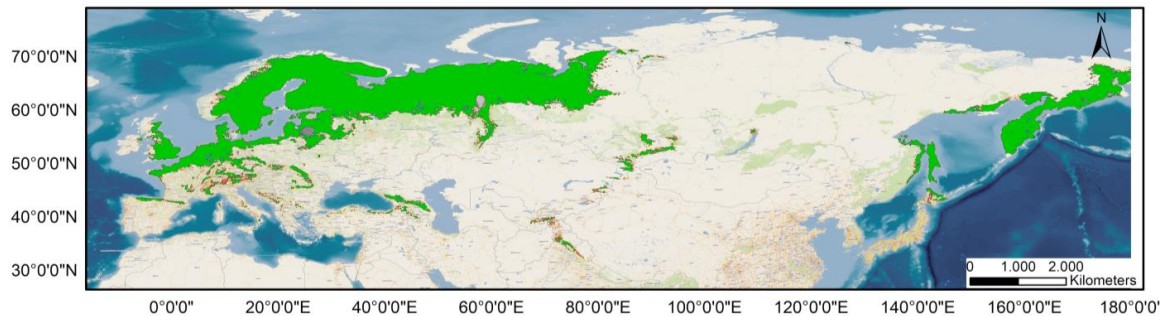

2080-2100 SSP 370

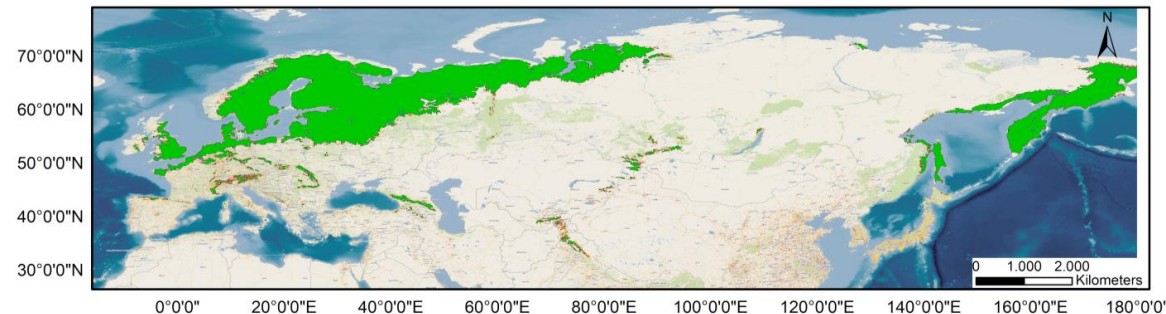

2021-2040 SSP 585

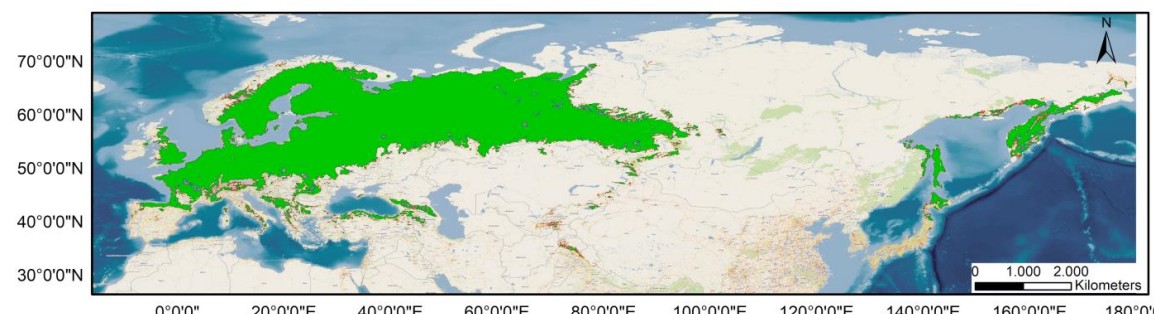

2041-2060 SSP 585

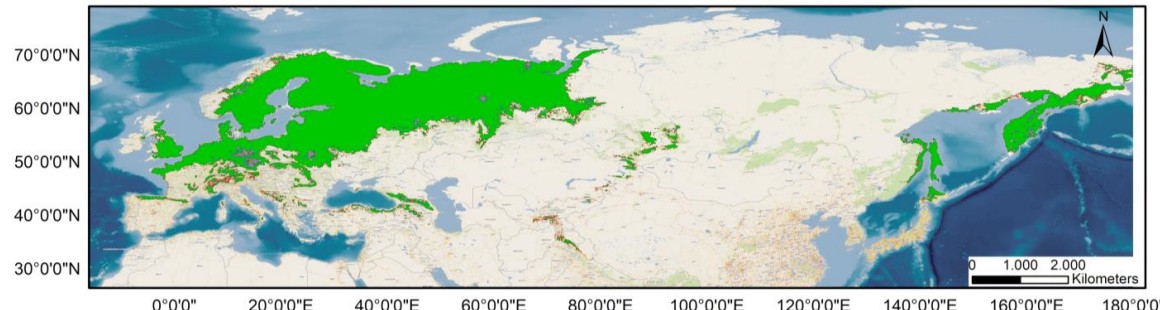

2061-2080 SSP 585

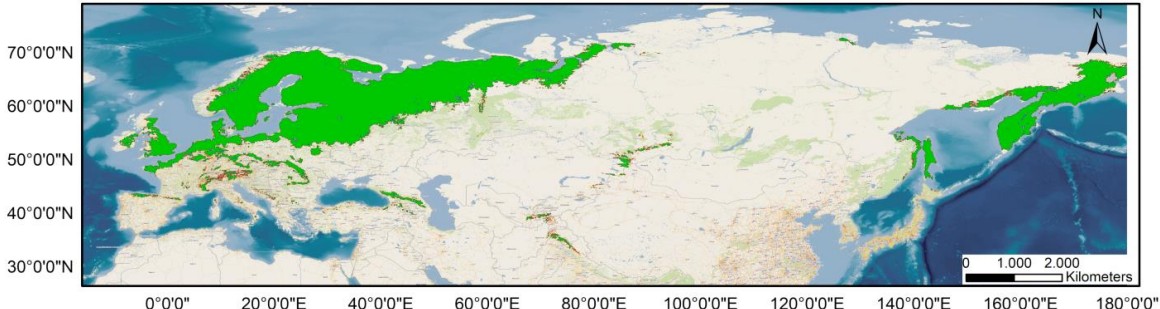

2080-2100 SSP 585

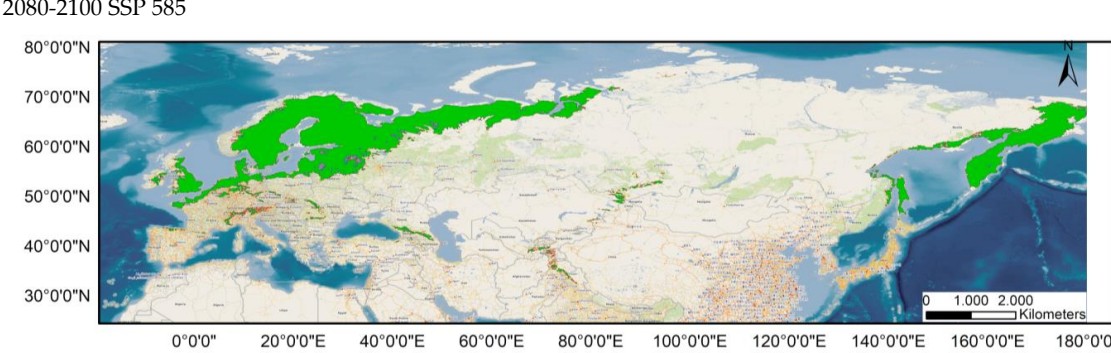

*Appendix A.3. Critical Areas for Connectivity under Different Climate Scenarios (60–100%)*

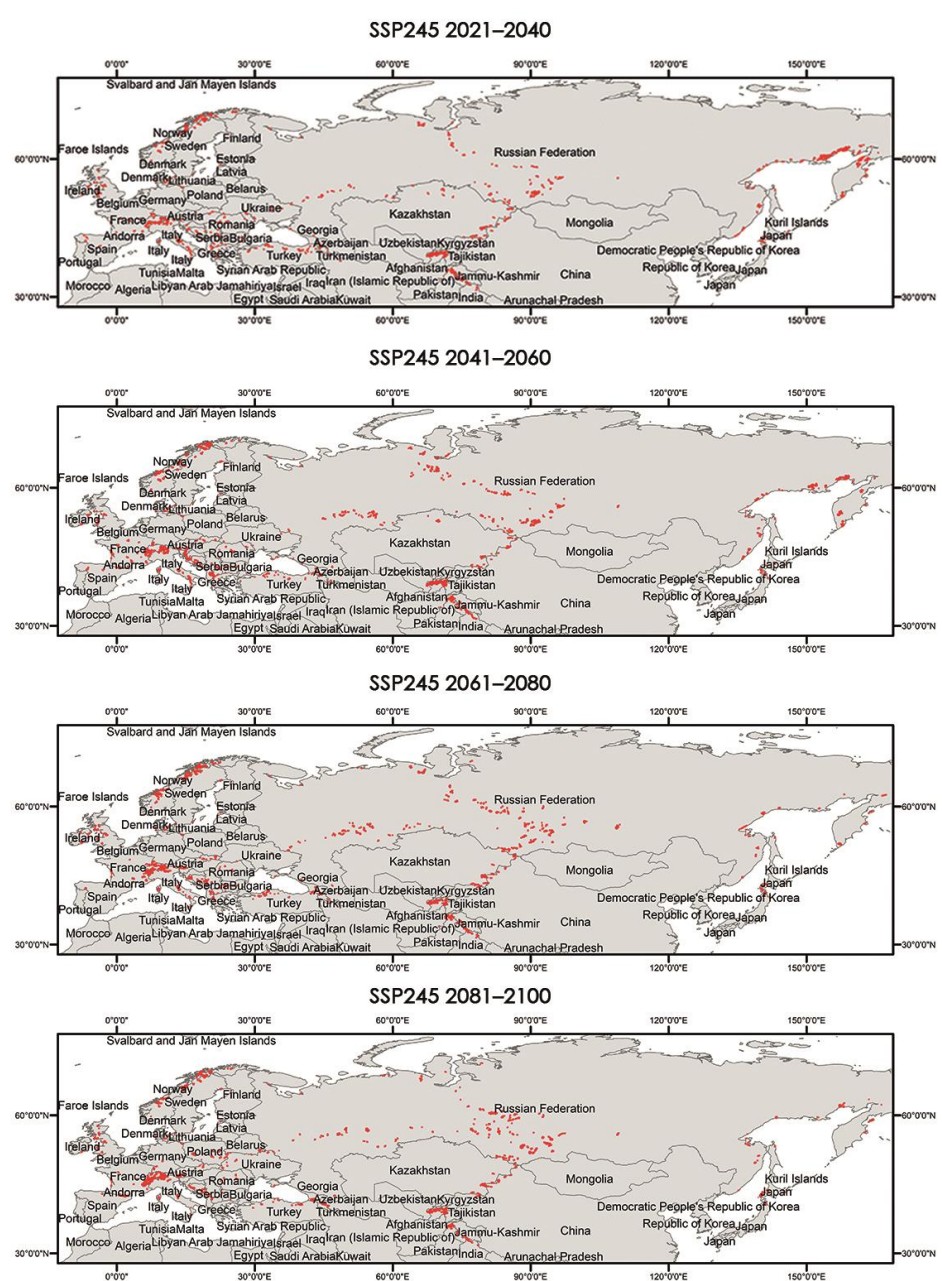

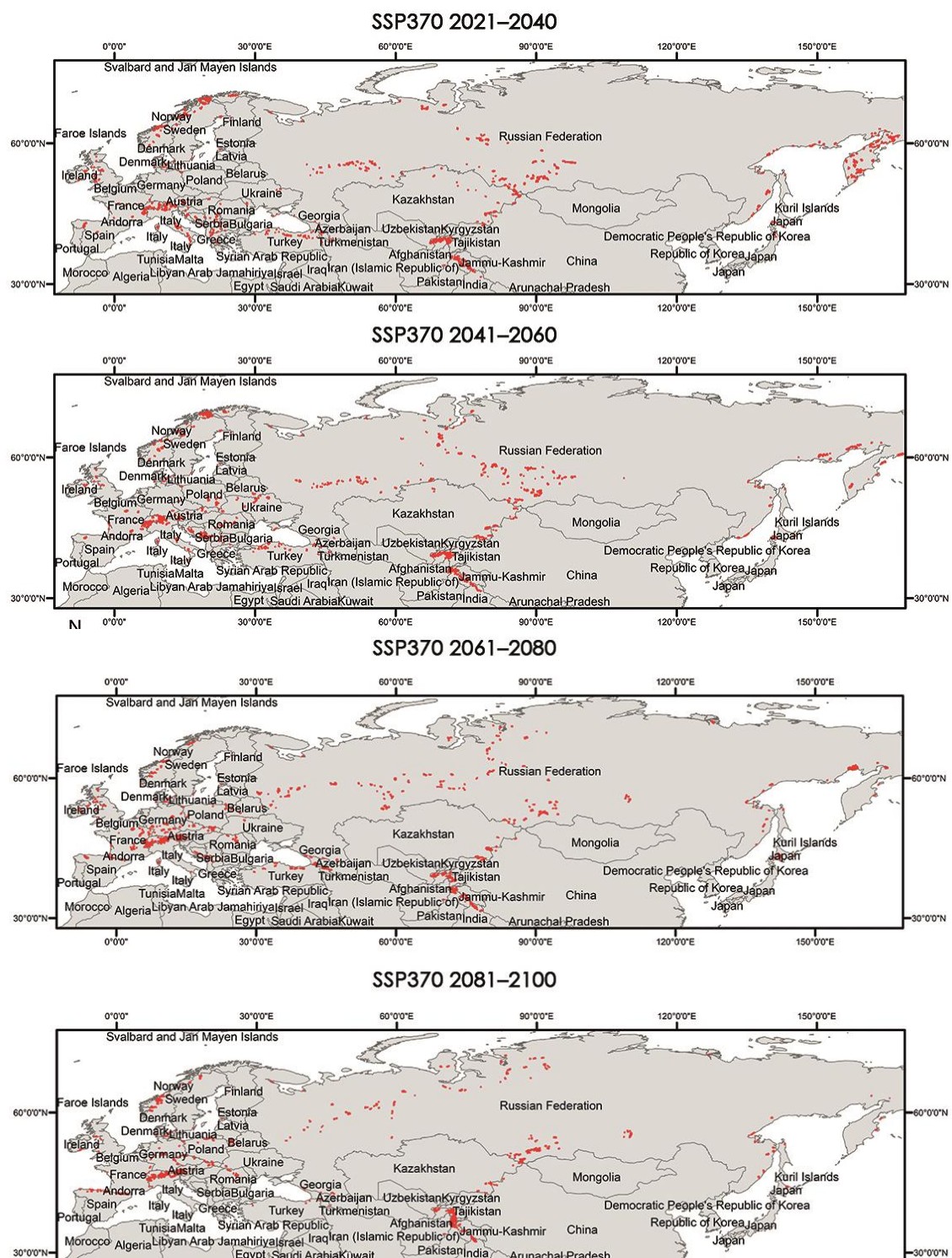

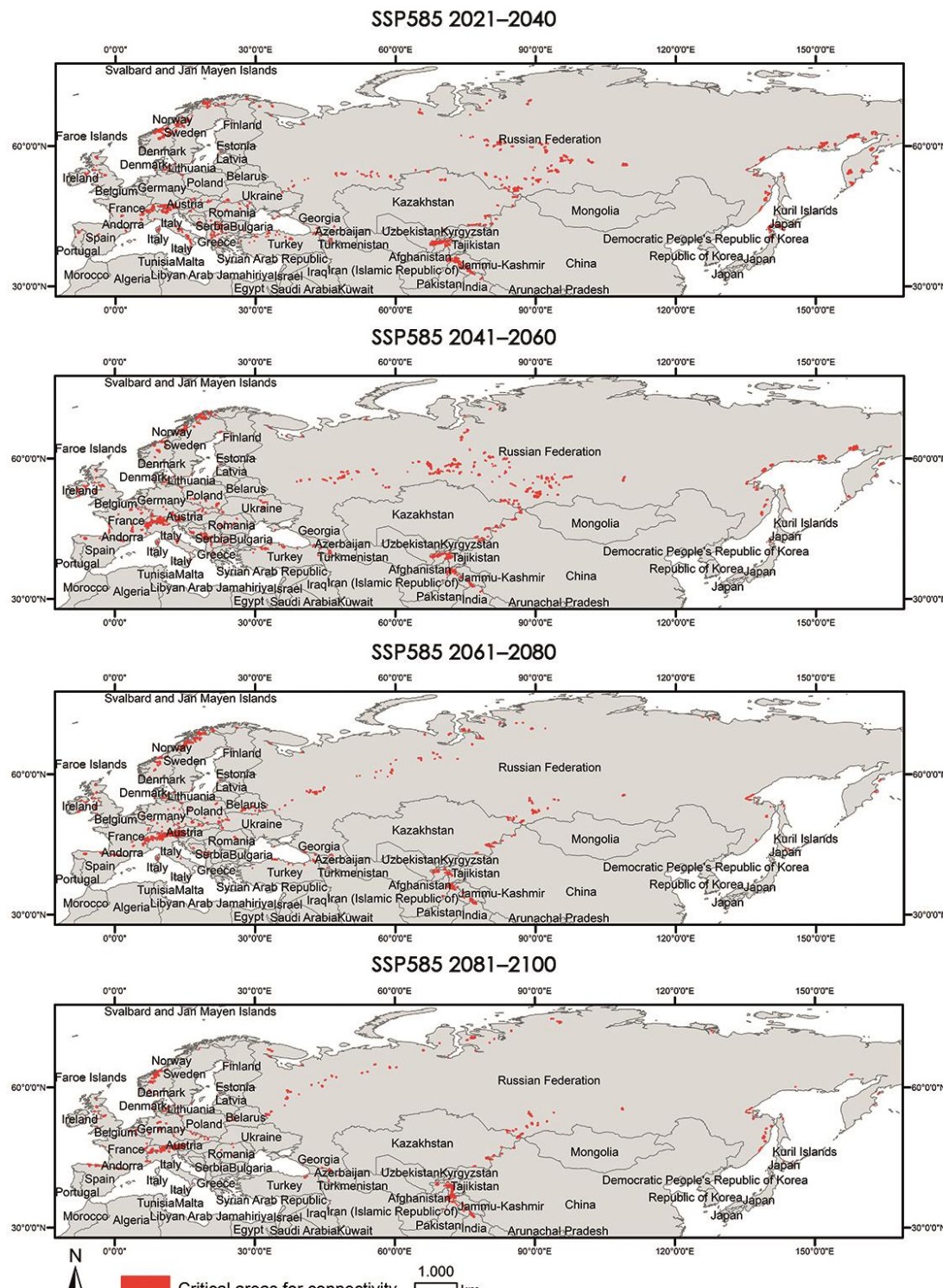

Figure: SSP585 2021–2040, SSP585 2041–2060, SSP585 2061–2080, SSP585 2081–2100.

N

Critical areas for connectivity

1.000 km

*Appendix A.4. Critical Areas for Connectivity under Different Climate Scenarios (80–100%)*

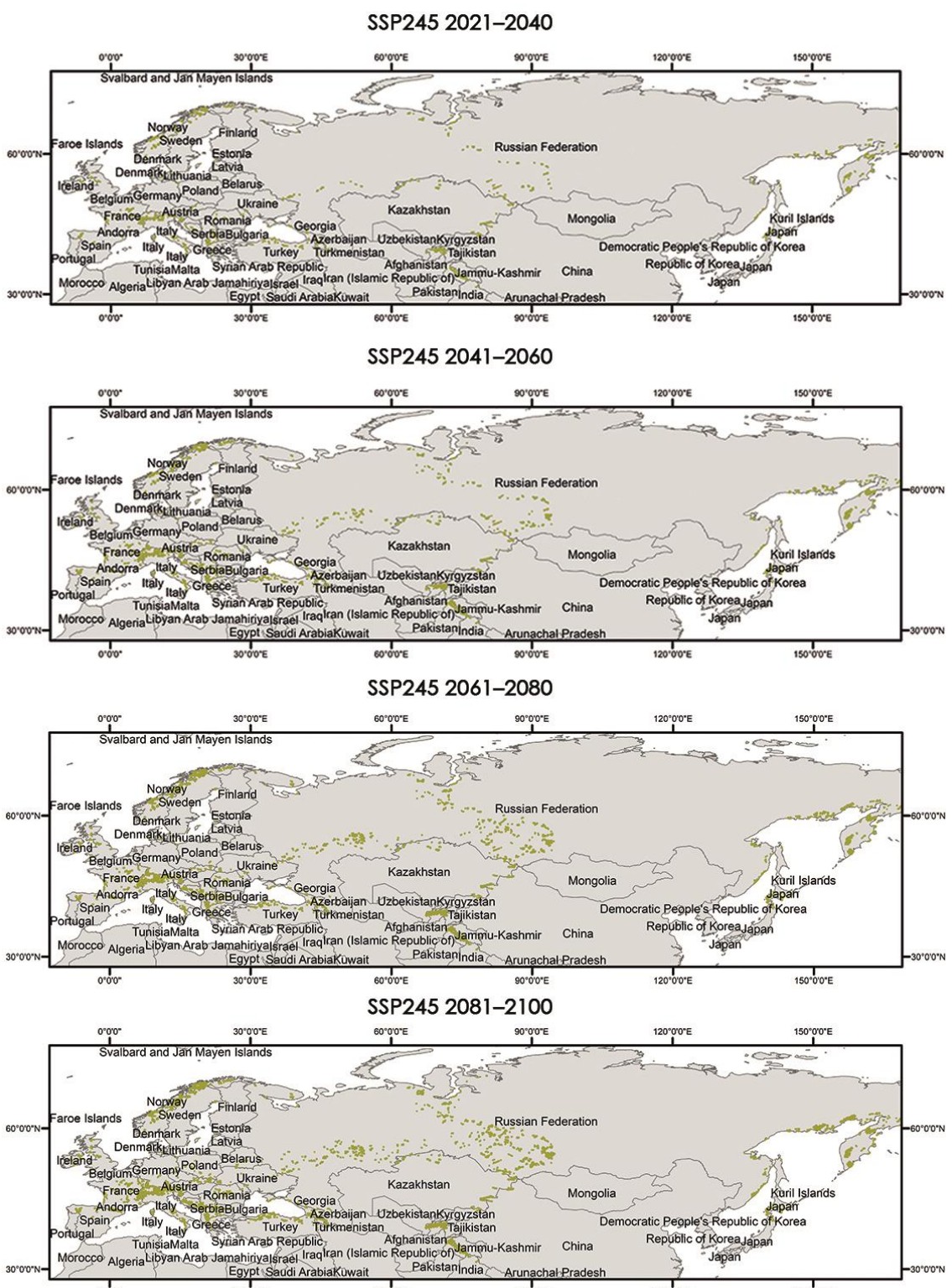

## SSP370 2021–2040

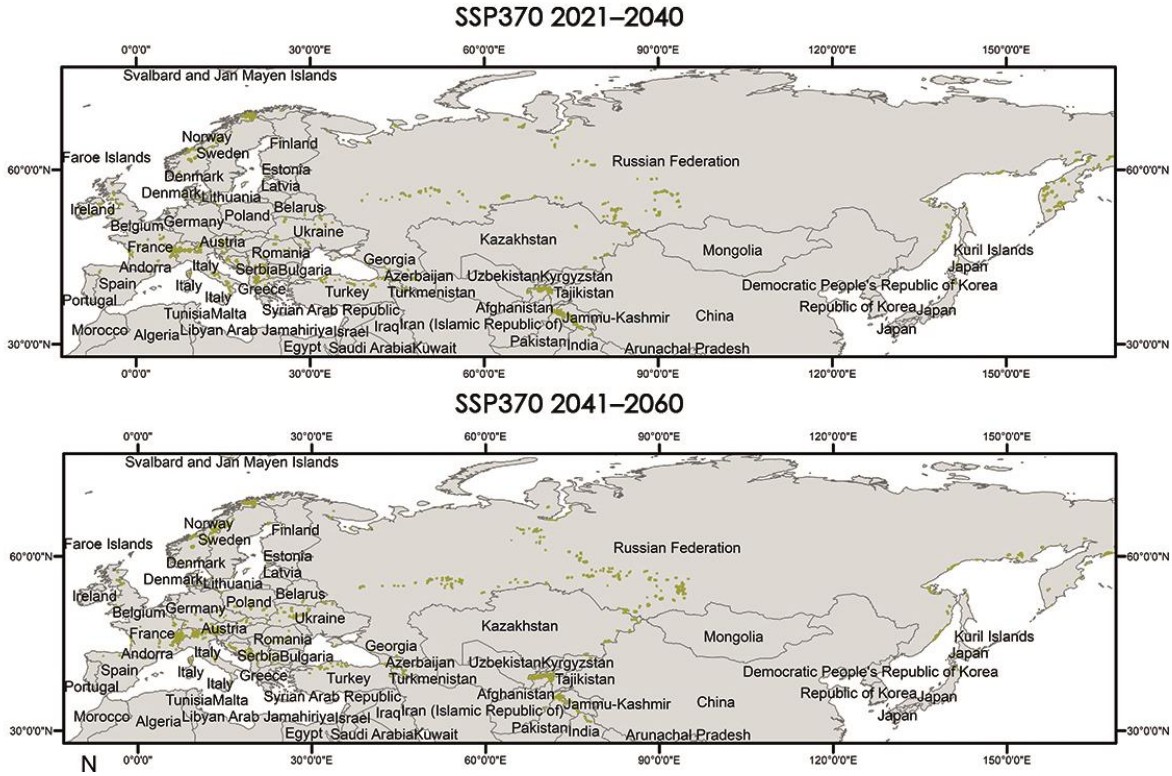

## SSP370 2041–2060

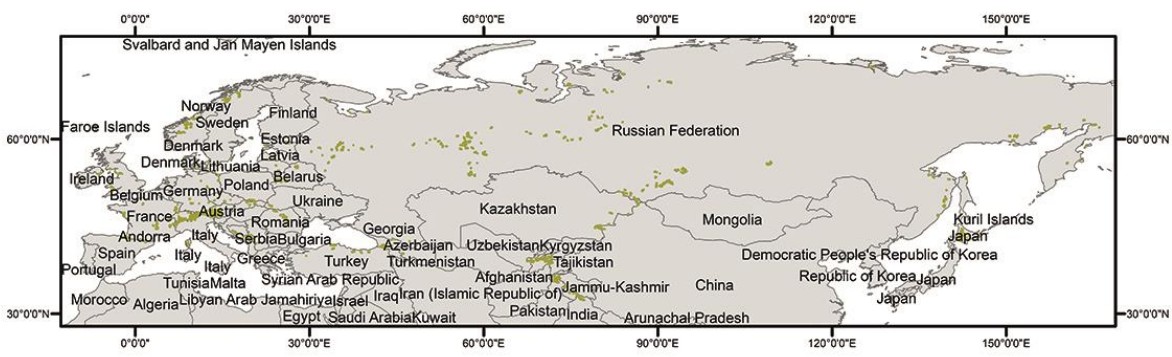

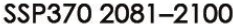

## SSP370 2061–2080

## SSP370 2081–2100

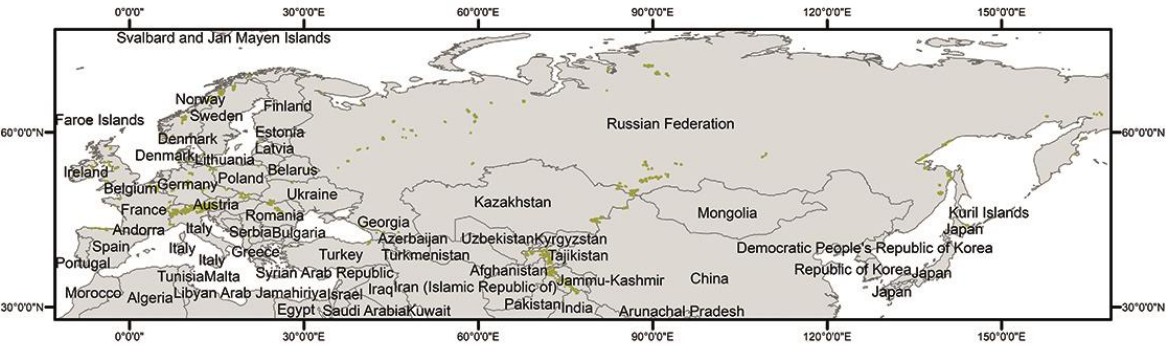

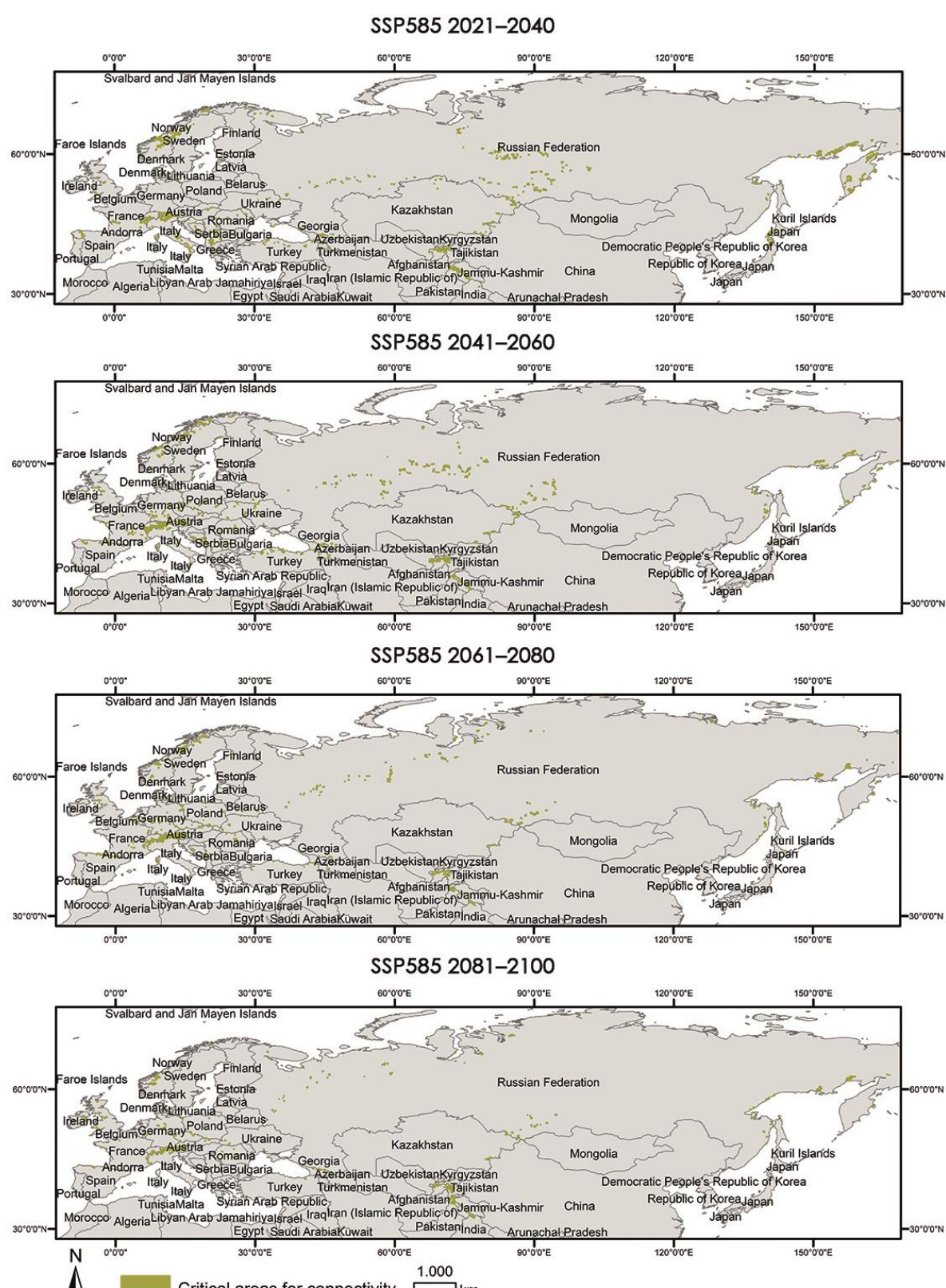

*Appendix A.5. MSPA Analysis under Different Climate Scenarios (60–100%)*

2021-2040 ssp 245

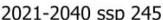

2041-2060 ssp 245

**PC value Cluster 1**
- 0 - 0,000000042
- 0,000000042 - 0,0000351
- 0,00003 - 0,0019
- 0,0019 - 10,4815

**PC value Cluster 2**
- 0 - 0,0000017
- 0,0000017 - 0,0000161
- 0,0000161 - 0,0002276
- 0,0002276 - 3,0076193

2061-2080 ssp 245

**PC value Cluster 3**
- 0 - 0,0020147
- 0,0020147 - 0,0039574
- 0,0039574 - 0,1125649
- 0,1125649 - 0,8347983

**PC value Cluster 4**
- 0 - 0,0676
- 0,0676 - 0,1353
- 0,1353 - 0,2029
- 0,2029 - 0,2706

2081-2100 ssp 245

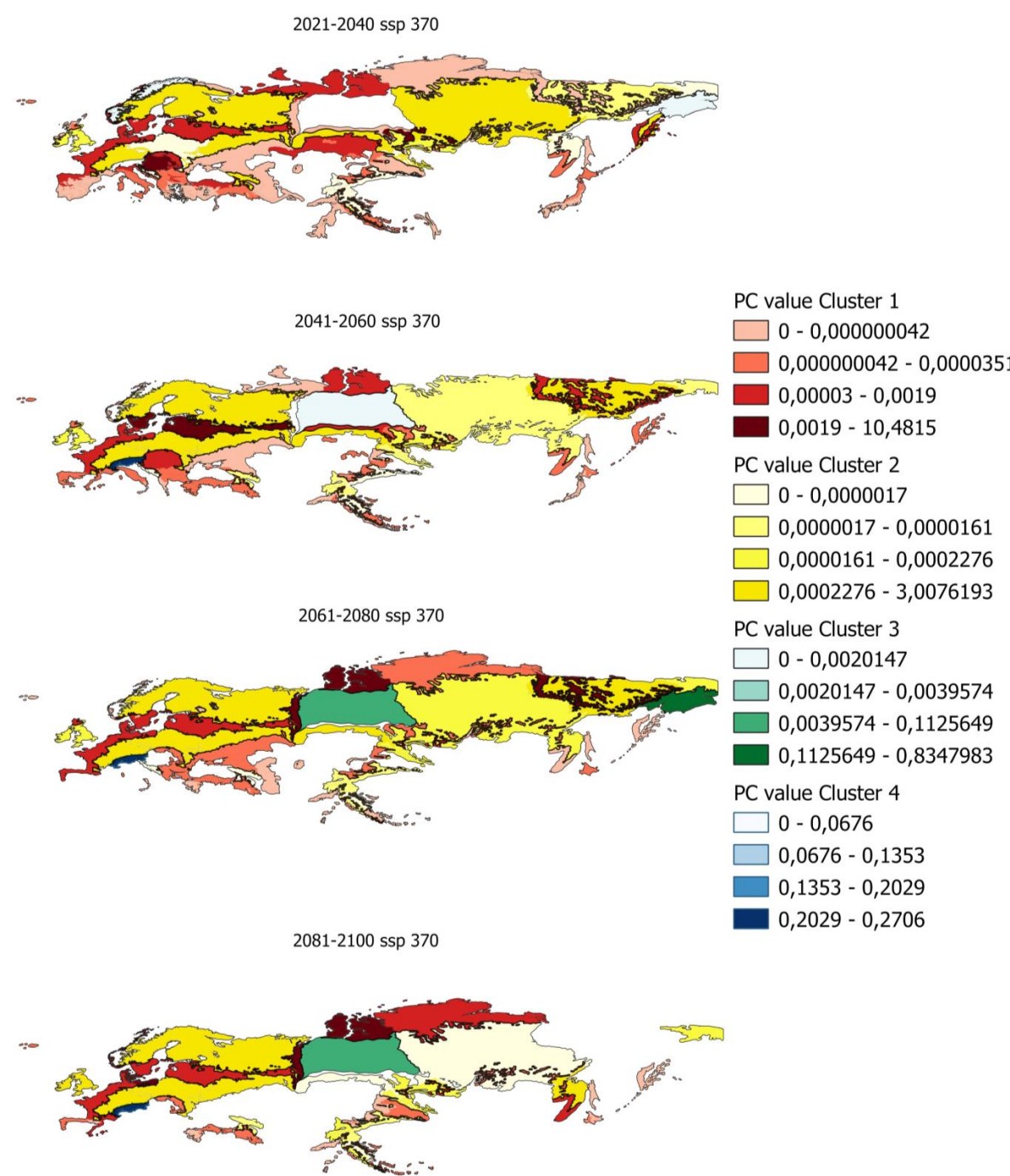

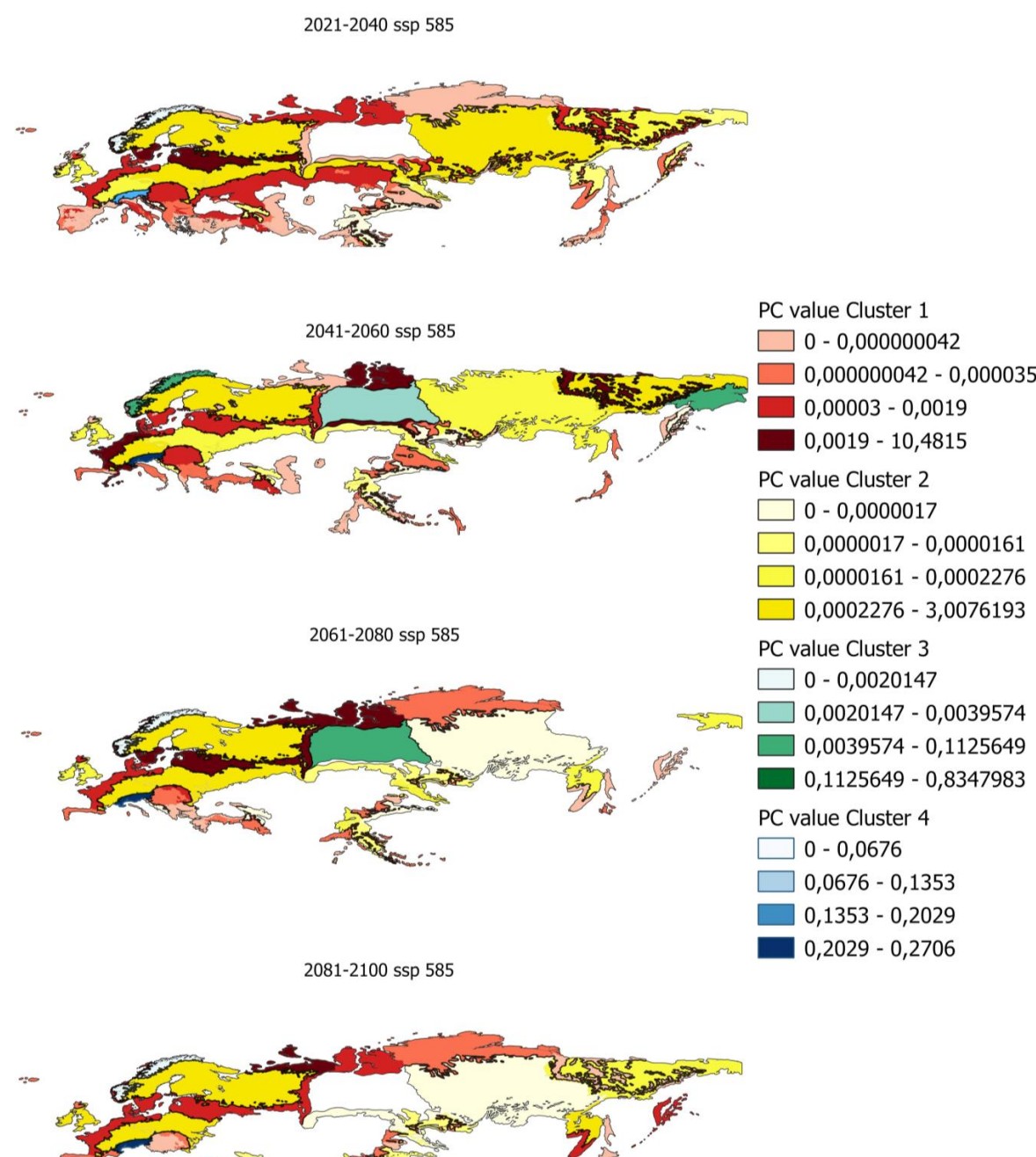

*Appendix A.6. MSPA Analysis under Different Climate Scenarios (80–100%)*

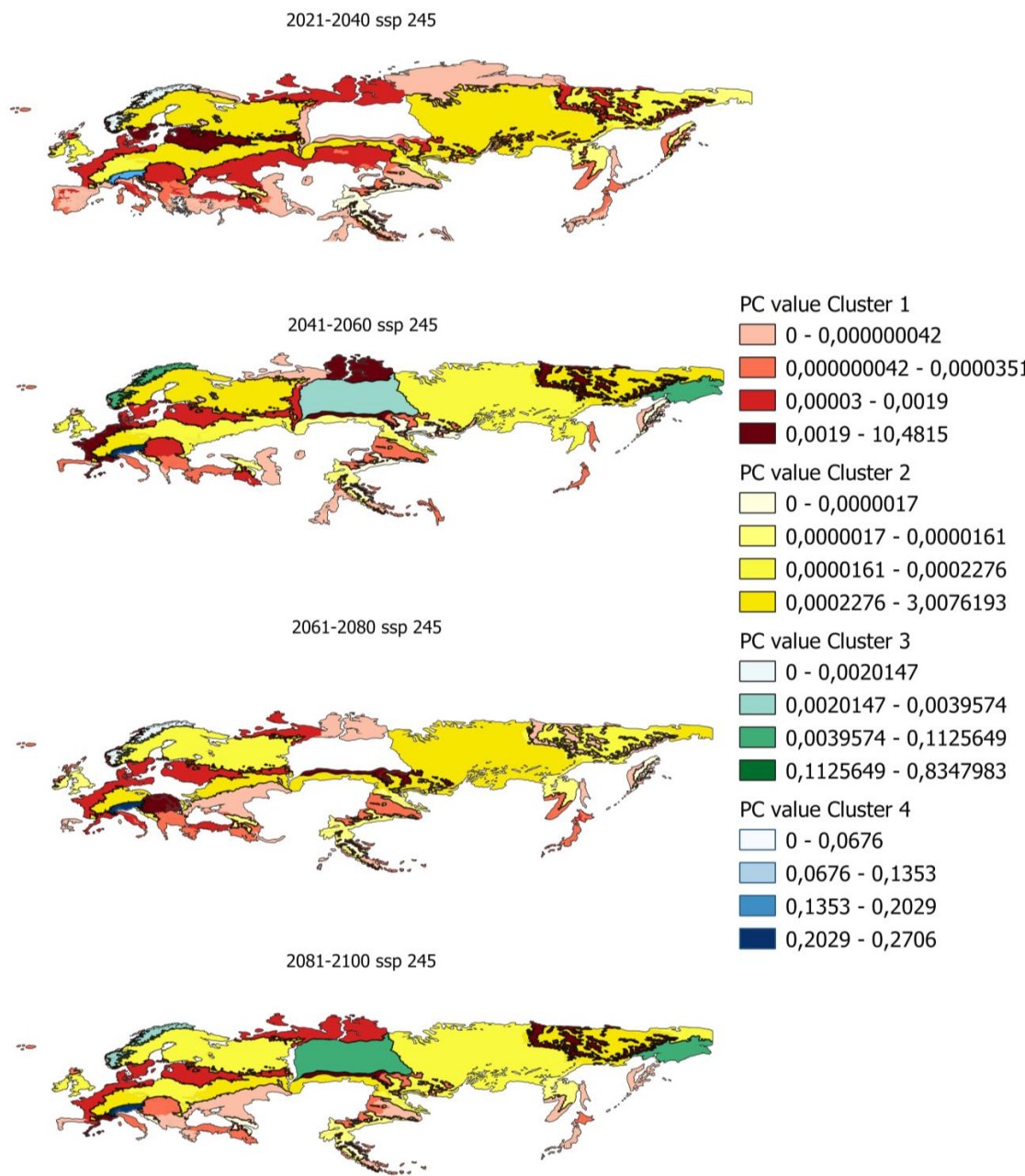

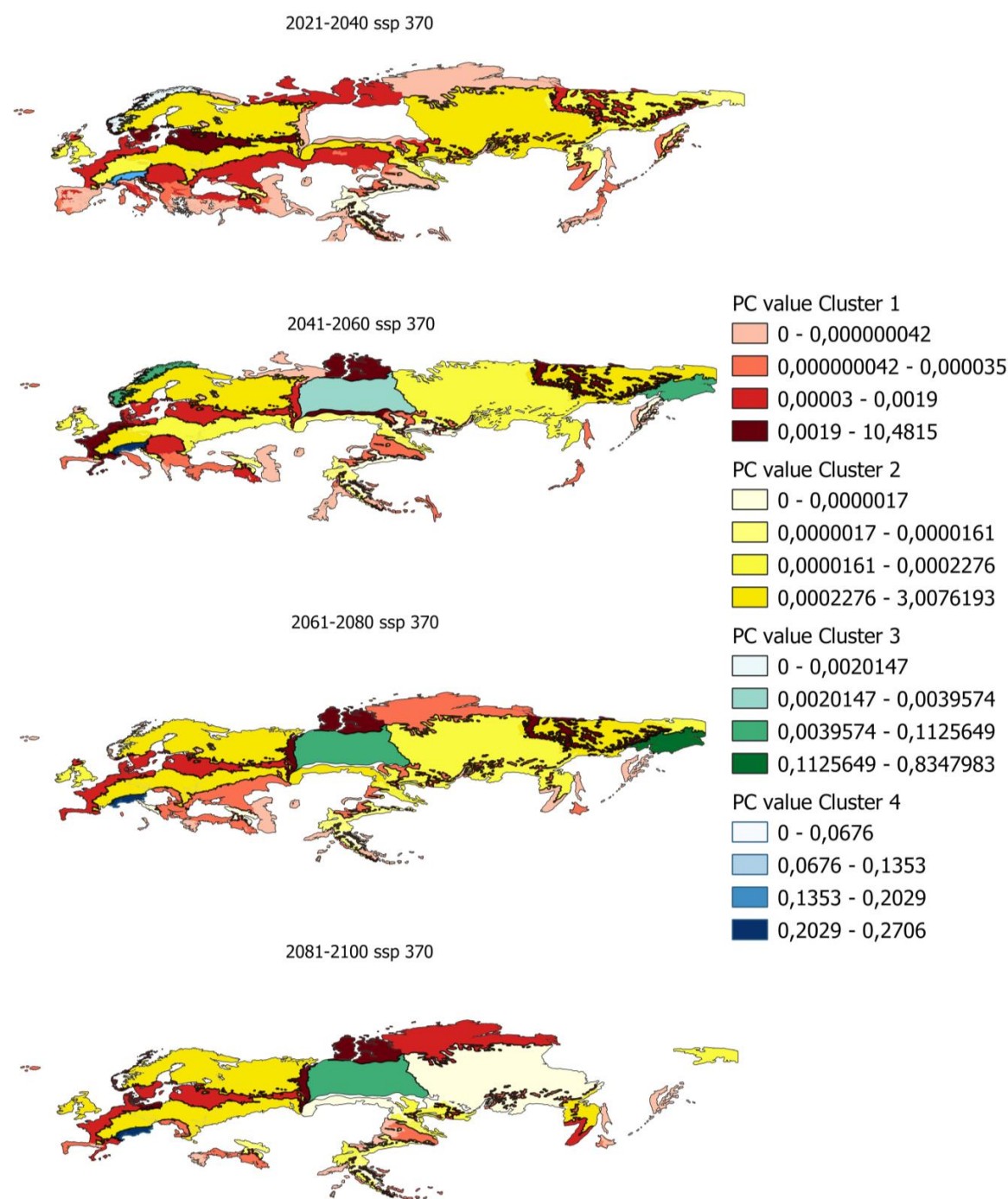

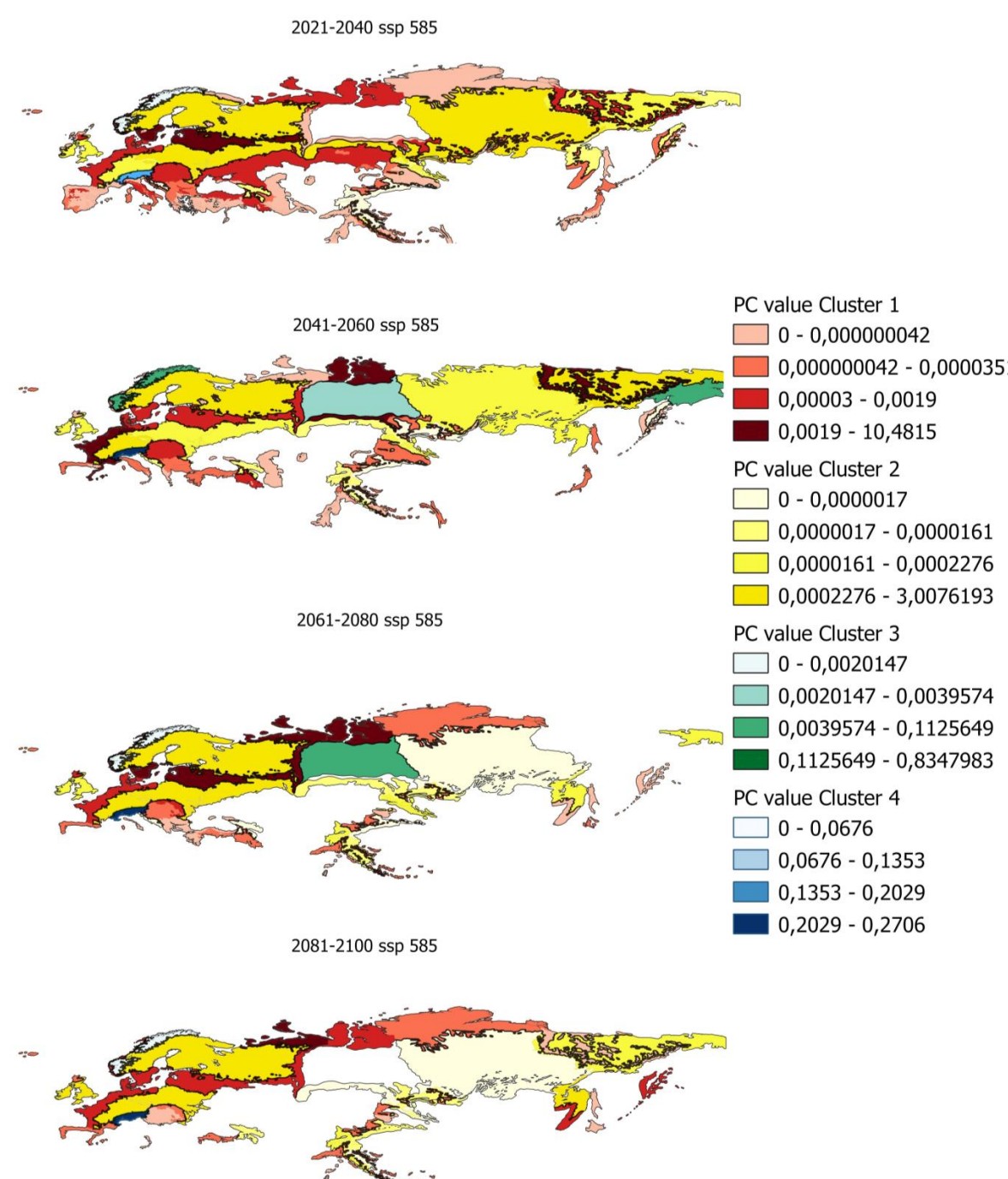

*Appendix A.7. Cluster Calculation (60–100%)*

| ECO_NAME | Terrestrial Ecoregion Clusters | C1_2020 | C245_2040 | C245_2060 | C245_2080 | C245_2100 | C370_2040 | C370_2060 | C370_2080 | C370_2100 | C585_2040 | C585_2060 | C585_2080 | C585_2100 |
|---|---|---|---|---|---|---|---|---|---|---|---|---|---|---|
| Aegean And Western Turkey Sclerophyllous And Mixed Forests | 1 | 53 | 3 | 5 | 0 | 0 | 2 | 0 | 0 | 0 | 2 | 0 | 0 | 0 |
| Alps Conifer And Mixed Forests | 4 | 0 | 466 | 601 | 674 | 836 | 671 | 747 | 867 | 832 | 526 | 812 | 949 | 810 |
| Altai Alpine Meadow And Tundra | 2 | 168 | 80 | 112 | 60 | 41 | 155 | 51 | 27 | 93 | 61 | 83 | 84 | 85 |
| Altai Montane Forest And Forest Steppe | 2 | 132 | 33 | 80 | 61 | 88 | 71 | 64 | 48 | 60 | 105 | 78 | 46 | 35 |
| Altai Steppe And Semi-Desert | 1 | 40 | 13 | 0 | 13 | 14 | 16 | 2 | 3 | 13 | 3 | 10 | 0 | 7 |
| Anatolian Conifer And Deciduous Mixed Forests | 1 | 124 | 25 | 2 | 4 | 8 | 0 | 4 | 10 | 2 | 9 | 4 | 2 | 0 |
| Appenine Deciduous Montane Forests | 1 | 5 | 57 | 39 | 44 | 25 | 60 | 22 | 0 | 0 | 47 | 28 | 5 | 0 |
| Atlantic Mixed Forests | 1 | 10 | 38 | 95 | 26 | 29 | 38 | 58 | 77 | 8 | 20 | 75 | 16 | 55 |
| Azerbaijan Shrub Desert And Steppe | 1 | 0 | 0 | 5 | 2 | 0 | 0 | 0 | 0 | 0 | 4 | 0 | 0 | 0 |
| Balkan Mixed Forests | 1 | 78 | 119 | 77 | 59 | 22 | 113 | 51 | 26 | 0 | 121 | 46 | 10 | 0 |
| Baltic Mixed Forests | 1 | 31 | 43 | 38 | 38 | 36 | 37 | 37 | 38 | 54 | 35 | 35 | 39 | 37 |
| Baluchistan Xeric Woodlands | 1 | 3 | 0 | 0 | 0 | 0 | 0 | 0 | 0 | 0 | 2 | 2 | 0 | 0 |
| Bering Tundra | 3 | 158 | 527 | 526 | 289 | 174 | 369 | 584 | 250 | 37 | 619 | 446 | 55 | 48 |
| Caledon Conifer Forests | 1 | 34 | 9 | 0 | 0 | 10 | 4 | 11 | 9 | 0 | 23 | 9 | 12 | 5 |
| Cantabrian Mixed Forests | 1 | 0 | 20 | 0 | 0 | 6 | 20 | 28 | 97 | 128 | 12 | 58 | 88 | 83 |
| Carpathian Montane Forests | 1 | 0 | 6 | 16 | 18 | 84 | 1 | 102 | 157 | 138 | 5 | 91 | 77 | 44 |
| Caspian Hyrcanian Mixed Forests | 1 | 0 | 6 | 0 | 2 | 0 | 0 | 0 | 0 | 0 | 0 | 0 | 0 | 0 |
| Caucasus Mixed Forests | 2 | 183 | 79 | 87 | 106 | 101 | 53 | 39 | 159 | 35 | 63 | 46 | 71 | 60 |
| Celtic Broadleaf Forests | 2 | 79 | 195 | 149 | 199 | 175 | 187 | 75 | 158 | 60 | 96 | 186 | 117 | 83 |
| Central Anatolian Steppe And Woodlands | 1 | 23 | 0 | 0 | 0 | 0 | 0 | 0 | 0 | 0 | 0 | 0 | 0 | 0 |
| Central European Mixed Forests | 2 | 0 | 22 | 20 | 0 | 225 | 25 | 211 | 197 | 146 | 62 | 88 | 219 | 43 |
| Cherskii-Kolyma Mountain Tundra | 1 | 31 | 32 | 135 | 4 | 16 | 0 | 122 | 30 | 0 | 90 | 103 | 0 | 17 |
| Chukchi Peninsula Tundra | 2 | 0 | 0 | 42 | 101 | 160 | 0 | 224 | 85 | 162 | 136 | 94 | 205 | 57 |
| Corsican Montane Broadleaf And Mixed Forests | 1 | 0 | 32 | 40 | 35 | 31 | 32 | 31 | 36 | 0 | 28 | 31 | 24 | 0 |
| Crimean Submediterranean Forest Complex | 1 | 6 | 5 | 19 | 9 | 8 | 13 | 9 | 0 | 0 | 16 | 4 | 3 | 0 |
| Dinaric Mountains Mixed Forests | 2 | 23 | 61 | 121 | 158 | 166 | 55 | 190 | 107 | 0 | 10 | 155 | 52 | 0 |
| East European Forest Steppe | 2 | 1 | 89 | 252 | 279 | 120 | 229 | 271 | 40 | 15 | 129 | 128 | 22 | 0 |
| East Siberian Taiga | 2 | 150 | 107 | 157 | 329 | 212 | 137 | 60 | 145 | 97 | 452 | 229 | 48 | 35 |
| Eastern Anatolian Deciduous Forests | 1 | 1 | 26 | 0 | 0 | 0 | 37 | 0 | 0 | 0 | 0 | 0 | 0 | 0 |
| Eastern Anatolian Montane Steppe | 1 | 84 | 34 | 18 | 12 | 6 | 44 | 28 | 8 | 14 | 15 | 13 | 4 | 0 |
| Elburz Range Forest Steppe | 1 | 0 | 6 | 0 | 0 | 0 | 0 | 0 | 0 | 0 | 0 | 0 | 0 | 0 |
| Emin Valley Steppe | 1 | 0 | 39 | 18 | 39 | 55 | 0 | 1 | 0 | 11 | 0 | 30 | 14 | 18 |
| Euxine-Colchic Broadleaf Forests | 1 | 8 | 62 | 41 | 69 | 44 | 60 | 99 | 63 | 0 | 56 | 64 | 29 | 0 |

| ECO_NAME | Terrestrial Ecoregion Clusters | C1_2020 | C245_2040 | C245_2060 | C245_2080 | C245_2100 | C370_2040 | C370_2060 | C370_2080 | C370_2100 | C585_2040 | C585_2060 | C585_2080 | C585_2100 |
|---|---|---|---|---|---|---|---|---|---|---|---|---|---|---|
| Gissaro-Alai Open Woodlands | 2 | 278 | 327 | 263 | 201 | 175 | 339 | 237 | 118 | 92 | 323 | 199 | 75 | 46 |
| Himalayan Subtropical Pine Forests | 1 | 19 | 8 | 12 | 0 | 0 | 8 | 9 | 0 | 0 | 9 | 0 | 0 | 0 |
| Hindu Kush Alpine Meadow | 1 | 0 | 0 | 0 | 0 | 0 | 0 | 15 | 0 | 0 | 25 | 13 | 15 | 11 |
| Hokkaido Deciduous Forests | 1 | 3 | 64 | 2 | 31 | 25 | 25 | 33 | 37 | 21 | 60 | 0 | 15 | 8 |
| Hokkaido Montane Conifer Forests | 1 | 23 | 13 | 2 | 5 | 12 | 3 | 26 | 47 | 3 | 36 | 0 | 0 | 8 |
| Honshu Alpine Conifer Forests | 1 | 7 | 4 | 9 | 8 | 16 | 0 | 0 | 0 | 0 | 19 | 3 | 0 | 0 |
| Iberian Conifer Forests | 1 | 7 | 21 | 0 | 0 | 0 | 22 | 0 | 0 | 0 | 18 | 0 | 0 | 0 |
| Iberian Sclerophyllous And Semi-Deciduous Forests | 1 | 36 | 0 | 0 | 0 | 0 | 0 | 0 | 0 | 0 | 0 | 0 | 0 | 0 |
| Illyrian Deciduous Forests | 1 | 123 | 11 | 22 | 0 | 0 | 21 | 0 | 0 | 0 | 5 | 0 | 0 | 0 |
| Italian Sclerophyllous And Semi-Deciduous Forests | 1 | 62 | 43 | 5 | 12 | 13 | 36 | 17 | 0 | 0 | 50 | 15 | 0 | 0 |
| Junggar Basin Semi-Desert | 1 | 23 | 0 | 0 | 4 | 5 | 8 | 0 | 0 | 4 | 0 | 32 | 0 | 0 |
| Kamchatka Mountain Tundra And Forest Tundra | 2 | 123 | 339 | 46 | 31 | 0 | 359 | 0 | 0 | 0 | 44 | 10 | 0 | 0 |
| Kamchatka-Kurile Meadows And Sparse Forests | 1 | 24 | 64 | 32 | 30 | 34 | 246 | 42 | 10 | 21 | 74 | 38 | 12 | 18 |
| Kamchatka-Kurile Taiga | 1 | 7 | 0 | 0 | 0 | 0 | 5 | 0 | 0 | 0 | 0 | 0 | 0 | 0 |
| Karakoram-West Tibetan Plateau Alpine Steppe | 2 | 11 | 50 | 134 | 155 | 99 | 42 | 129 | 156 | 115 | 52 | 96 | 103 | 78 |
| Kazakh Forest Steppe | 2 | 146 | 68 | 46 | 84 | 86 | 96 | 142 | 17 | 23 | 144 | 45 | 7 | 13 |
| Kazakh Upland | 1 | 52 | 3 | 44 | 0 | 0 | 22 | 15 | 0 | 0 | 48 | 0 | 0 | 0 |
| Kola Peninsula Tundra | 1 | 0 | 0 | 0 | 0 | 0 | 0 | 0 | 0 | 0 | 6 | 0 | 0 | 0 |
| Kopet Dag Woodlands And Forest Steppe | 1 | 4 | 0 | 0 | 0 | 0 | 0 | 0 | 0 | 0 | 0 | 0 | 0 | 0 |
| Lake: Palearctic | 1 | 8 | 0 | 0 | 0 | 0 | 0 | 0 | 1 | 0 | 0 | 6 | 0 | 0 |
| Nihonkai Montane Deciduous Forests | 1 | 110 | 46 | 42 | 52 | 28 | 24 | 1 | 0 | 0 | 74 | 14 | 0 | 0 |
| North Atlantic Moist Mixed Forests | 1 | 0 | 0 | 0 | 12 | 0 | 0 | 0 | 0 | 0 | 0 | 0 | 0 | 0 |
| Northeast Siberian Taiga | 2 | 21 | 383 | 118 | 23 | 36 | 103 | 86 | 7 | 0 | 181 | 89 | 0 | 4 |
| Northeastern Spain And Southern France Mediterranean Forests | 1 | 25 | 15 | 47 | 41 | 17 | 0 | 8 | 0 | 1 | 30 | 14 | 0 | 5 |
| Northern Anatolian Conifer And Deciduous Forests | 1 | 82 | 50 | 49 | 53 | 47 | 71 | 65 | 53 | 8 | 36 | 46 | 18 | 21 |
| Northwest Iberian Montane Forests | 1 | 32 | 14 | 44 | 21 | 0 | 48 | 3 | 0 | 0 | 12 | 0 | 0 | 0 |
| Northwest Russian-Novaya Zemlya Tundra | 1 | 35 | 86 | 74 | 138 | 71 | 95 | 8 | 0 | 0 | 41 | 9 | 15 | 37 |
| Northwestern Himalayan Alpine Shrub And Meadows | 2 | 248 | 236 | 188 | 126 | 86 | 217 | 223 | 154 | 102 | 271 | 141 | 142 | 96 |
| Nujiang Langcang Gorge Alpine Conifer And Mixed Forests | 1 | 0 | 0 | 0 | 0 | 0 | 0 | 0 | 0 | 0 | 0 | 10 | 0 | 0 |
| Okhotsk-Manchurian Taiga | 2 | 44 | 128 | 108 | 97 | 61 | 131 | 45 | 103 | 94 | 199 | 220 | 85 | 165 |
| Pamir Alpine Desert And Tundra | 2 | 108 | 202 | 241 | 205 | 193 | 225 | 244 | 227 | 275 | 169 | 216 | 94 | 188 |
| Pannonian Mixed Forests | 1 | 0 | 110 | 126 | 78 | 15 | 56 | 53 | 0 | 0 | 68 | 52 | 18 | 1 |
| Paropamisus Xeric Woodlands | 1 | 0 | 0 | 0 | 0 | 0 | 0 | 4 | 11 | 27 | 0 | 0 | 11 | 28 |

| ECO_NAME | Terrestrial Ecoregion Clusters | C1_2020 | C245_2040 | C245_2060 | C245_2080 | C245_2100 | C370_2040 | C370_2060 | C370_2080 | C370_2100 | C585_2040 | C585_2060 | C585_2080 | C585_2100 |
|---|---|---|---|---|---|---|---|---|---|---|---|---|---|---|
| Pindus Mountains Mixed Forests | 1 | 23 | 60 | 66 | 74 | 0 | 77 | 9 | 0 | 0 | 51 | 35 | 0 | 0 |
| Po Basin Mixed Forests | 1 | 0 | 1 | 0 | 0 | 0 | 1 | 0 | 0 | 0 | 0 | 0 | 0 | 0 |
| Pontic Steppe | 1 | 163 | 93 | 27 | 4 | 13 | 0 | 4 | 16 | 0 | 12 | 0 | 0 | 0 |
| Pyrenees Conifer And Mixed Forests | 1 | 4 | 0 | 40 | 0 | 82 | 0 | 0 | 34 | 149 | 0 | 24 | 48 | 55 |
| Rock And Ice: Palearctic | 1 | 2 | 10 | 16 | 41 | 37 | 10 | 40 | 51 | 113 | 6 | 45 | 51 | 147 |
| Rodope Montane Mixed Forests | 1 | 0 | 23 | 39 | 50 | 10 | 22 | 27 | 0 | 0 | 34 | 37 | 0 | 0 |
| Sakhalin Island Taiga | 1 | 36 | 4 | 4 | 4 | 4 | 4 | 4 | 4 | 4 | 4 | 5 | 4 | 4 |
| Sarmatic Mixed Forests | 1 | 3 | 3 | 10 | 72 | 91 | 117 | 42 | 166 | 20 | 3 | 139 | 104 | 93 |
| Sayan Alpine Meadows And Tundra | 1 | 55 | 35 | 39 | 10 | 39 | 55 | 48 | 11 | 45 | 14 | 48 | 45 | 10 |
| Sayan Montane Conifer Forests | 2 | 350 | 115 | 187 | 112 | 245 | 171 | 131 | 201 | 188 | 67 | 181 | 127 | 115 |
| Scandinavian And Russian Taiga | 2 | 286 | 55 | 125 | 59 | 82 | 93 | 101 | 214 | 191 | 367 | 115 | 86 | 224 |
| Scandinavian Coastal Conifer Forests | 1 | 0 | 40 | 50 | 27 | 2 | 37 | 6 | 10 | 13 | 54 | 2 | 25 | 27 |
| Scandinavian Montane Birch Forest And Grasslands | 3 | 106 | 521 | 460 | 664 | 495 | 563 | 538 | 149 | 240 | 447 | 491 | 609 | 226 |
| South Appenine Mixed Montane Forests | 1 | 2 | 27 | 43 | 25 | 12 | 27 | 27 | 6 | 0 | 27 | 14 | 0 | 2 |
| South Sakhalin-Kurile Mixed Forests | 1 | 4 | 8 | 0 | 0 | 0 | 0 | 0 | 0 | 0 | 0 | 0 | 0 | 0 |
| South Siberian Forest Steppe | 1 | 37 | 82 | 84 | 119 | 29 | 93 | 138 | 9 | 0 | 50 | 97 | 0 | 0 |
| Southern Anatolian Montane Conifer And Deciduous Forests | 1 | 70 | 0 | 0 | 0 | 0 | 0 | 0 | 0 | 0 | 0 | 0 | 0 | 0 |
| Southwest Iberian Mediterranean Sclerophyllous And Mixed Forests | 1 | 0 | 0 | 0 | 0 | 0 | 0 | 0 | 0 | 0 | 3 | 0 | 0 | 0 |
| Sulaiman Range Alpine Meadows | 1 | 26 | 2 | 10 | 2 | 0 | 11 | 0 | 4 | 7 | 0 | 2 | 10 | 5 |
| Taiheiyo Evergreen Forests | 1 | 2 | 0 | 0 | 0 | 0 | 0 | 0 | 0 | 0 | 0 | 0 | 0 | 0 |
| Taiheiyo Montane Deciduous Forests | 1 | 16 | 0 | 5 | 4 | 1 | 0 | 0 | 0 | 0 | 0 | 0 | 0 | 0 |
| Taimyr-Central Siberian Tundra | 1 | 0 | 0 | 0 | 0 | 0 | 0 | 0 | 115 | 131 | 47 | 0 | 61 | 88 |
| Tian Shan Foothill Arid Steppe | 1 | 92 | 67 | 41 | 49 | 41 | 36 | 24 | 58 | 23 | 87 | 20 | 38 | 16 |
| Tian Shan Montane Conifer Forests | 1 | 52 | 19 | 24 | 38 | 55 | 26 | 40 | 55 | 8 | 26 | 42 | 8 | 0 |
| Tian Shan Montane Steppe And Meadows | 2 | 53 | 84 | 47 | 145 | 101 | 89 | 129 | 139 | 51 | 63 | 88 | 99 | 62 |
| Trans-Baikal Bald Mountain Tundra | 1 | 5 | 0 | 0 | 0 | 0 | 0 | 0 | 32 | 60 | 3 | 0 | 0 | 0 |
| Tyrrhenian-Adriatic Sclerophyllous And Mixed Forests | 1 | 13 | 43 | 51 | 0 | 0 | 43 | 0 | 0 | 0 | 41 | 0 | 0 | 0 |
| Ural Montane Forests And Tundra | 1 | 0 | 0 | 1 | 0 | 0 | 0 | 0 | 84 | 77 | 0 | 11 | 36 | 6 |
| Ussuri Broadleaf And Mixed Forests | 1 | 54 | 56 | 70 | 20 | 6 | 66 | 62 | 7 | 30 | 22 | 0 | 15 | 28 |

| ECO_NAME | Terrestrial Ecoregion Clusters | C1_2020 | C245_2040 | C245_2060 | C245_2080 | C245_2100 | C370_2040 | C370_2060 | C370_2080 | C370_2100 | C585_2040 | C585_2060 | C585_2080 | C585_2100 |
|---|---|---|---|---|---|---|---|---|---|---|---|---|---|---|
| West Siberian Taiga | 3 | 294 | 195 | 448 | 398 | 391 | 195 | 451 | 360 | 109 | 207 | 547 | 91 | 12 |
| Western European Broadleaf Forests | 2 | 0 | 90 | 112 | 178 | 157 | 66 | 125 | 470 | 179 | 73 | 221 | 171 | 155 |
| Western Himalayan Alpine Shrub And Meadows | 1 | 7 | 3 | 2 | 4 | 4 | 3 | 2 | 18 | 0 | 6 | 4 | 0 | 0 |
| Western Himalayan Broadleaf Forests | 2 | 151 | 100 | 87 | 73 | 61 | 163 | 104 | 85 | 62 | 177 | 52 | 52 | 42 |
| Western Himalayan Subalpine Conifer Forests | 1 | 70 | 30 | 54 | 36 | 24 | 42 | 53 | 42 | 40 | 40 | 44 | 36 | 34 |
| Western Siberian Hemiboreal Forests | 1 | 4 | 4 | 0 | 19 | 101 | 7 | 85 | 0 | 0 | 0 | 183 | 0 | 0 |
| Yamal-Gydan Tundra | 1 | 21 | 9 | 31 | 12 | 15 | 35 | 60 | 139 | 116 | 37 | 51 | 143 | 105 |
| Kazakh Steppe | 1 | 40 | 132 | 61 | 0 | 0 | 64 | 0 | 0 | 0 | 35 | 0 | 0 | 0 |

*Appendix A.8. Cluster Calculation (80–100%)*

| ECO_NAME | Terrestrial Ecoregions Clusters | C1_2020 | C245_2040 | C245_2060 | C245_2080 | C245_2100 | C370_2040 | C370_2060 | C370_2080 | C370_2100 | C585_2040 | C585_2060 | C585_2080 | C585_2100 |
|---|---|---|---|---|---|---|---|---|---|---|---|---|---|---|
| Aegean And Western Turkey Sclerophyllous And Mixed Forests | 1 | 50 | 2 | 4 | 0 | 0 | 2 | 0 | 0 | 0 | 2 | 0 | 0 | 0 |
| Alps Conifer And Mixed Forests | 4 | 512 | 585 | 663 | 647 | 769 | 548 | 852 | 822 | 899 | 638 | 959 | 1042 | 849 |
| Altai Alpine Meadow And Tundra | 2 | 174 | 56 | 52 | 115 | 85 | 72 | 57 | 71 | 129 | 96 | 114 | 80 | 41 |
| Altai Montane Forest And Forest Steppe | 1 | 105 | 41 | 54 | 24 | 48 | 38 | 75 | 71 | 65 | 60 | 48 | 46 | 62 |
| Altai Steppe And Semi-Desert | 1 | 28 | 20 | 4 | 17 | 14 | 24 | 14 | 11 | 6 | 18 | 11 | 0 | 7 |
| Anatolian Conifer And Deciduous Mixed Forests | 1 | 100 | 0 | 0 | 4 | 8 | 0 | 8 | 2 | 0 | 0 | 4 | 6 | 0 |
| Appenine Deciduous Montane Forests | 1 | 9 | 51 | 45 | 44 | 3 | 57 | 26 | 0 | 0 | 59 | 7 | 0 | 0 |
| Atlantic Mixed Forests | 1 | 10 | 47 | 26 | 56 | 43 | 46 | 23 | 43 | 86 | 63 | 20 | 58 | 39 |
| Azerbaijan Shrub Desert And Steppe | 1 | 0 | 0 | 5 | 5 | 0 | 2 | 0 | 0 | 0 | 0 | 0 | 0 | 0 |
| Balkan Mixed Forests | 1 | 53 | 102 | 67 | 53 | 31 | 107 | 34 | 21 | 2 | 101 | 21 | 5 | 0 |
| Baltic Mixed Forests | 1 | 13 | 26 | 25 | 13 | 33 | 15 | 15 | 38 | 39 | 27 | 13 | 47 | 16 |
| Bering Tundra | 3 | 136 | 343 | 306 | 281 | 465 | 292 | 656 | 130 | 64 | 497 | 514 | 214 | 82 |
| Caledon Conifer Forests | 1 | 14 | 0 | 10 | 5 | 6 | 1 | 9 | 0 | 15 | 0 | 0 | 5 | 8 |
| Cantabrian Mixed Forests | 1 | 0 | 24 | 25 | 8 | 18 | 1 | 8 | 25 | 55 | 70 | 106 | 113 | 39 |
| Carpathian Montane Forests | 1 | 0 | 1 | 35 | 52 | 92 | 6 | 84 | 178 | 180 | 7 | 89 | 106 | 79 |
| Caspian Hyrcanian Mixed Forests | 1 | 20 | 0 | 0 | 0 | 0 | 0 | 0 | 0 | 0 | 0 | 0 | 0 | 0 |

| ECO_NAME | Terrestrial Ecoregions Clusters | C1_2020 | C245_2040 | C245_2060 | C245_2080 | C245_2100 | C370_2040 | C370_2060 | C370_2080 | C370_2100 | C585_2040 | C585_2060 | C585_2080 | C585_2100 |
|---|---|---|---|---|---|---|---|---|---|---|---|---|---|---|
| Caucasus Mixed Forests | 2 | 329 | 141 | 88 | 53 | 127 | 101 | 88 | 75 | 27 | 168 | 146 | 49 | 90 |
| Celtic Broadleaf Forests | 2 | 90 | 64 | 131 | 85 | 117 | 165 | 67 | 87 | 109 | 64 | 71 | 116 | 166 |
| Central European Mixed Forests | 2 | 15 | 40 | 0 | 15 | 155 | 19 | 378 | 113 | 23 | 40 | 232 | 93 | 10 |
| Cherskii-Kolyma Mountain Tundra | 1 | 0 | 12 | 60 | 64 | 101 | 7 | 67 | 65 | 21 | 82 | 152 | 126 | 99 |
| Chukchi Peninsula Tundra | 1 | 0 | 0 | 39 | 157 | 70 | 0 | 112 | 119 | 202 | 16 | 146 | 228 | 45 |
| Corsican Montane Broadleaf And Mixed Forests | 1 | 0 | 28 | 36 | 35 | 33 | 28 | 35 | 26 | 0 | 26 | 33 | 23 | 0 |
| Crimean Submediterranean Forest Complex | 1 | 0 | 36 | 9 | 6 | 9 | 7 | 5 | 0 | 0 | 23 | 6 | 0 | 0 |
| Dinaric Mountains Mixed Forests | 2 | 70 | 31 | 170 | 148 | 128 | 77 | 153 | 75 | 0 | 25 | 138 | 77 | 0 |
| East European Forest Steppe | 2 | 20 | 148 | 284 | 263 | 78 | 138 | 169 | 7 | 15 | 221 | 125 | 26 | 0 |
| East Siberian Taiga | 2 | 77 | 42 | 137 | 131 | 159 | 50 | 233 | 177 | 49 | 304 | 58 | 7 | 29 |
| Eastern Anatolian Deciduous Forests | 1 | 10 | 32 | 0 | 0 | 0 | 30 | 0 | 0 | 0 | 14 | 0 | 0 | 0 |
| Eastern Anatolian Montane Steppe | 1 | 117 | 38 | 9 | 12 | 7 | 29 | 15 | 2 | 0 | 26 | 18 | 4 | 9 |
| Emin Valley Steppe | 1 | 0 | 0 | 0 | 8 | 0 | 0 | 0 | 0 | 0 | 0 | 0 | 0 | 0 |
| Euxine-Colchic Broadleaf Forests | 1 | 19 | 79 | 59 | 76 | 58 | 72 | 73 | 43 | 0 | 71 | 98 | 5 | 0 |
| Gissaro-Alai Open Woodlands | 2 | 152 | 184 | 192 | 226 | 162 | 220 | 251 | 111 | 72 | 190 | 195 | 79 | 76 |
| Himalayan Subtropical Pine Forests | 1 | 22 | 8 | 5 | 8 | 0 | 0 | 1 | 0 | 0 | 0 | 0 | 0 | 0 |
| Hokkaido Deciduous Forests | 1 | 2 | 35 | 81 | 20 | 10 | 12 | 12 | 121 | 20 | 50 | 30 | 10 | 0 |
| Hokkaido Montane Conifer Forests | 1 | 20 | 15 | 54 | 4 | 12 | 7 | 4 | 91 | 0 | 40 | 23 | 0 | 0 |
| Honshu Alpine Conifer Forests | 1 | 12 | 12 | 4 | 16 | 0 | 0 | 0 | 0 | 0 | 16 | 0 | 0 | 0 |
| Iberian Conifer Forests | 1 | 0 | 19 | 0 | 0 | 0 | 0 | 0 | 0 | 0 | 0 | 0 | 0 | 0 |
| Iberian Sclerophyllous And Semi-Deciduous Forests | 1 | 49 | 0 | 0 | 0 | 0 | 0 | 0 | 0 | 0 | 0 | 0 | 0 | 0 |
| Illyrian Deciduous Forests | 1 | 45 | 0 | 0 | 0 | 0 | 6 | 0 | 0 | 0 | 18 | 0 | 0 | 0 |
| Italian Sclerophyllous And Semi-Deciduous Forests | 1 | 105 | 22 | 19 | 18 | 0 | 17 | 13 | 0 | 0 | 53 | 5 | 1 | 0 |
| Junggar Basin Semi-Desert | 1 | 14 | 0 | 0 | 0 | 0 | 9 | 5 | 4 | 0 | 11 | 5 | 0 | 0 |
| Kamchatka Mountain Tundra And Forest Tundra | 2 | 122 | 373 | 28 | 0 | 0 | 87 | 5 | 0 | 0 | 494 | 34 | 0 | 0 |
| Kamchatka-Kurile Meadows And Sparse Forests | 2 | 10 | 169 | 58 | 31 | 14 | 314 | 21 | 15 | 16 | 202 | 60 | 24 | 8 |
| Kamchatka-Kurile Taiga | 1 | 0 | 4 | 0 | 0 | 0 | 0 | 0 | 0 | 0 | 6 | 0 | 0 | 0 |
| Karakoram-West Tibetan Plateau Alpine Steppe | 1 | 16 | 1 | 122 | 56 | 105 | 32 | 98 | 94 | 87 | 19 | 50 | 65 | 80 |

| ECO_NAME | Terrestrial Ecoregions Clusters | C1_2020 | C245_2040 | C245_2060 | C245_2080 | C245_2100 | C370_2040 | C370_2060 | C370_2080 | C370_2100 | C585_2040 | C585_2060 | C585_2080 | C585_2100 |
|---|---|---|---|---|---|---|---|---|---|---|---|---|---|---|
| Kazakh Forest Steppe | 2 | 112 | 80 | 90 | 70 | 24 | 173 | 75 | 16 | 9 | 132 | 21 | 16 | 1 |
| Kazakh Upland | 1 | 25 | 32 | 17 | 0 | 0 | 0 | 0 | 0 | 0 | 9 | 0 | 0 | 0 |
| Kola Peninsula Tundra | 1 | 13 | 15 | 19 | 12 | 0 | 0 | 0 | 0 | 0 | 0 | 0 | 0 | 0 |
| Lake: Palearctic | 1 | 7 | 0 | 0 | 0 | 0 | 0 | 1 | 0 | 0 | 0 | 0 | 0 | 0 |
| Nihonkai Montane Deciduous Forests | 1 | 100 | 34 | 21 | 45 | 6 | 20 | 6 | 0 | 0 | 95 | 8 | 0 | 0 |
| Northeast Siberian Taiga | 2 | 14 | 175 | 146 | 64 | 101 | 39 | 92 | 33 | 0 | 373 | 102 | 115 | 56 |
| Northeastern Spain And Southern France Mediterranean Forests | 1 | 24 | 2 | 43 | 4 | 8 | 2 | 10 | 6 | 4 | 17 | 13 | 2 | 4 |
| Northern Anatolian Conifer And Deciduous Forests | 1 | 35 | 46 | 50 | 56 | 52 | 41 | 56 | 39 | 8 | 30 | 55 | 10 | 12 |
| Northwest Iberian Montane Forests | 1 | 62 | 60 | 46 | 0 | 0 | 11 | 0 | 0 | 0 | 22 | 0 | 0 | 0 |
| Northwest Russian-Novaya Zemlya Tundra | 1 | 0 | 69 | 77 | 81 | 85 | 101 | 2 | 0 | 0 | 0 | 0 | 0 | 0 |
| Northwestern Himalayan Alpine Shrub And Meadows | 2 | 214 | 113 | 174 | 90 | 74 | 278 | 173 | 119 | 133 | 227 | 90 | 103 | 110 |
| Okhotsk-Manchurian Taiga | 2 | 55 | 22 | 16 | 56 | 68 | 123 | 114 | 112 | 226 | 75 | 123 | 107 | 138 |
| Pamir Alpine Desert And Tundra | 2 | 93 | 137 | 238 | 252 | 226 | 160 | 272 | 199 | 225 | 150 | 251 | 149 | 174 |
| Pannonian Mixed Forests | 1 | 0 | 134 | 93 | 81 | 27 | 72 | 31 | 0 | 0 | 89 | 28 | 18 | 0 |
| Paropamisus Xeric Woodlands | 1 | 0 | 0 | 0 | 0 | 0 | 0 | 0 | 0 | 1 | 0 | 0 | 0 | 17 |
| Pindus Mountains Mixed Forests | 1 | 16 | 75 | 57 | 67 | 0 | 68 | 2 | 0 | 0 | 95 | 2 | 0 | 0 |
| Po Basin Mixed Forests | 1 | 5 | 2 | 0 | 0 | 0 | 2 | 0 | 0 | 0 | 0 | 0 | 0 | 0 |
| Pontic Steppe | 1 | 119 | 21 | 6 | 1 | 4 | 29 | 0 | 4 | 0 | 6 | 14 | 0 | 0 |
| Pyrenees Conifer And Mixed Forests | 1 | 0 | 0 | 99 | 5 | 195 | 0 | 2 | 29 | 0 | 0 | 26 | 59 | 2 |
| Rock And Ice: Palearctic | 1 | 0 | 10 | 9 | 26 | 40 | 10 | 20 | 39 | 105 | 0 | 24 | 29 | 138 |
| Rodope Montane Mixed Forests | 1 | 10 | 31 | 40 | 47 | 18 | 32 | 31 | 0 | 0 | 35 | 32 | 0 | 0 |
| Sakhalin Island Taiga | 1 | 51 | 5 | 4 | 3 | 4 | 4 | 4 | 4 | 4 | 4 | 3 | 4 | 4 |
| Sarmatic Mixed Forests | 1 | 4 | 4 | 35 | 75 | 45 | 123 | 162 | 150 | 29 | 4 | 71 | 37 | 21 |
| Sayan Alpine Meadows And Tundra | 1 | 0 | 10 | 12 | 19 | 46 | 19 | 13 | 35 | 41 | 14 | 21 | 46 | 31 |
| Sayan Montane Conifer Forests | 2 | 251 | 82 | 97 | 137 | 134 | 65 | 100 | 163 | 155 | 133 | 143 | 96 | 59 |
| Scandinavian And Russian Taiga | 2 | 394 | 127 | 156 | 110 | 73 | 102 | 248 | 310 | 116 | 486 | 142 | 148 | 169 |
| Scandinavian Coastal Conifer Forests | 1 | 0 | 47 | 43 | 4 | 22 | 26 | 33 | 2 | 5 | 35 | 22 | 0 | 0 |
| Scandinavian Montane Birch Forest And Grasslands | 3 | 84 | 494 | 518 | 538 | 496 | 449 | 420 | 282 | 255 | 324 | 400 | 478 | 242 |

| ECO_NAME | Terrestrial Ecoregions Clusters | C1_2020 | C245_2040 | C245_2060 | C245_2080 | C245_2100 | C370_2040 | C370_2060 | C370_2080 | C370_2100 | C585_2040 | C585_2060 | C585_2080 | C585_2100 |
|---|---|---|---|---|---|---|---|---|---|---|---|---|---|---|
| South Appenine Mixed Montane Forests | 1 | 17 | 51 | 39 | 23 | 16 | 54 | 12 | 0 | 0 | 41 | 17 | 0 | 0 |
| South Sakhalin-Kurile Mixed Forests | 1 | 2 | 0 | 0 | 0 | 0 | 0 | 0 | 0 | 0 | 0 | 0 | 0 | 0 |
| South Siberian Forest Steppe | 1 | 24 | 14 | 17 | 72 | 82 | 57 | 82 | 0 | 0 | 54 | 52 | 0 | 0 |
| Southern Anatolian Montane Conifer And Deciduous Forests | 1 | 24 | 0 | 0 | 0 | 0 | 0 | 0 | 0 | 0 | 0 | 0 | 0 | 0 |
| Sulaiman Range Alpine Meadows | 1 | 5 | 0 | 0 | 0 | 0 | 0 | 11 | 0 | 0 | 0 | 0 | 5 | 0 |
| Taiheiyo Evergreen Forests | 1 | 0 | 0 | 0 | 0 | 0 | 0 | 0 | 0 | 0 | 2 | 0 | 0 | 0 |
| Taiheiyo Montane Deciduous Forests | 1 | 34 | 0 | 0 | 5 | 5 | 0 | 3 | 0 | 0 | 0 | 0 | 0 | 0 |
| Taimyr-Central Siberian Tundra | 1 | 0 | 0 | 0 | 0 | 0 | 0 | 0 | 88 | 142 | 0 | 0 | 61 | 48 |
| Tian Shan Foothill Arid Steppe | 1 | 0 | 18 | 34 | 18 | 30 | 23 | 3 | 27 | 34 | 24 | 18 | 0 | 3 |
| Tian Shan Montane Conifer Forests | 1 | 10 | 58 | 30 | 12 | 0 | 11 | 11 | 0 | 0 | 30 | 20 | 0 | 0 |
| Tian Shan Montane Steppe And Meadows | 1 | 4 | 104 | 73 | 68 | 118 | 40 | 59 | 76 | 96 | 78 | 78 | 44 | 40 |
| Trans-Baikal Bald Mountain Tundra | 1 | 0 | 0 | 0 | 0 | 0 | 0 | 0 | 0 | 9 | 0 | 0 | 0 | 0 |
| Tyrrhenian-Adriatic Sclerophyllous And Mixed Forests | 1 | 27 | 43 | 10 | 0 | 0 | 41 | 0 | 0 | 0 | 43 | 0 | 0 | 0 |
| Ural Montane Forests And Tundra | 1 | 0 | 0 | 0 | 0 | 7 | 0 | 0 | 219 | 91 | 0 | 56 | 84 | 26 |
| Ussuri Broadleaf And Mixed Forests | 1 | 7 | 28 | 24 | 13 | 32 | 14 | 59 | 14 | 0 | 1 | 4 | 23 | 0 |
| West Siberian Taiga | 3 | 263 | 186 | 227 | 369 | 347 | 143 | 341 | 236 | 35 | 409 | 577 | 123 | 1 |
| Western European Broadleaf Forests | 2 | 30 | 113 | 203 | 146 | 234 | 147 | 226 | 254 | 129 | 38 | 258 | 288 | 158 |
| Western Himalayan Alpine Shrub And Meadows | 1 | 9 | 7 | 7 | 0 | 0 | 7 | 2 | 0 | 0 | 1 | 0 | 0 | 0 |
| Western Himalayan Broadleaf Forests | 2 | 169 | 167 | 85 | 113 | 56 | 139 | 63 | 70 | 70 | 132 | 66 | 31 | 28 |
| Western Himalayan Subalpine Conifer Forests | 1 | 69 | 26 | 32 | 40 | 26 | 39 | 39 | 41 | 41 | 33 | 36 | 40 | 29 |
| Western Siberian Hemiboreal Forests | 1 | 7 | 0 | 0 | 140 | 60 | 46 | 167 | 0 | 0 | 33 | 5 | 0 | 0 |
| Yamal-Gydan Tundra | 1 | 23 | 42 | 7 | 0 | 23 | 27 | 37 | 81 | 5 | 33 | 7 | 129 | 53 |
| Kazakh Steppe | 1 | 12 | 133 | 3 | 0 | 0 | 154 | 0 | 0 | 0 | 12 | 0 | 0 | 0 |

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
