# Peer review of "Mapping Priority Areas for Connectivity of Yellow-Winged Darter (Sympetrum flaveolum, Linnaeus 1758) under Climate Change"

_land, doi:10.3390/land12020298_

Round 1

Reviewer 1 Report

The manuscript is well prepared. Still, I have a few comments:

I have doubts about the limited ecology of the species. At least in central Europe, the species is considered as eurytopic, found not only on raised bogs or fens, but also on waterbodies, garden pools, wetland pools, oxbow lakes, quarry pools, and even fish ponds, and artificial canals.

L123-L124: During data thinning, how much data was discarded, and how much was retained for subsequent analysis? When we did a similar type of analysis on another species of dragonfly, we downloaded the data from GBIF and then eliminated it within a certain buffer, there was a large degree of data aggregation, so I'm interested (readers may be interested also) in how much of it was actually left for analysis with the selected buffer.

L134: Averages for precipitation and temperature are from 1970 to 2000 but what about data from GBIF? Are they limited to the same time frame, or are you working with all available data, even older ones? My concern is that climate change is already taking place during the period 1970-2000, and the records of the species from earlier years may not correspond to the conditions that prevailed here in the years 1970-2000.

L137-138: “We removed the variance inflation factor lower than 10” – maybe you meant to write that lower than 10 were kept, and higher than 10 were removed. In addition, 10 is quite a high number, usually, it is worked with a limit of 2.5 or 5, but I understand that when working with worldclime data, you need to choose a higher limit. The second option was to replace the mentioned worldclime variables with the axes of their PCA, as is standard with high collinearity in the data.

All images should be larger in size and resolution, although at the expense of their number. It shouldn't be a problem in the supplement.

Author Response

In the attached document we have tried to answer all the doubts.

Reviewer 2 Report

Comments and Suggestions for Authors

Dear authors!

I have important remarks about the text of your manuscript.

1. In introduction of the manuscript does not fully reflect the distribution of this species of dragonflies, as well as its ecology and behavior. It is necessary to describe these moments of the life of this species more precisely. It will be very interesting for potential readers of this article.

Therefore, it will be necessary to use and cite a number of articles devoted to these issues.

Sympetrum flaveolum is found in the temperate cold zone of Eurasia. It is usually confined to the mountains in the south of its range. But this species is common and abundant in a large part of eastern and central Europe and in the southern half of Fennoscandia.

This species is invasive throughout most of its range, especially in the lowlands. It may be absent from a certain area for a long time, breeding there for several years after the arrival of large flocks or individual migrants. For example, its distribution in the lowlands of western and southern Europe and in the south of eastern Europe is largely dependent on invasions from central and eastern Europe, which were usually associated with strong easterly winds. These invasions, such as in 1995 and 2006, can be significant and often result in the establishment of numerous temporary populations. But as a rule, the resulting lowland populations are short-lived and in most cases die out after a few years (up to 5-6 years). This can be thought of as an "influx model pattern" followed by decline and disappearance.

S. flaveolum occurs in a wide range of stagnant waters that are neither too eutrophic nor heavily shaded.

On the plain, the species (unstable temporary populations) prefers stagnant shallow water bodies, partially or completely drying up in summer: temporary water meadows, shallow dune lakes and ponds, small low-lying swamps and quarries.

While stable mountain populations are found on sphagnum peat bogs, small alkaline or acid lakes and temporary ponds.

Egg clutches are very resistant to drying out and withstand drying and severe winter frosts (up to - 30°-40°C). The development of larvae is fast, and generally depends on the temperature and the rate of drying of the reservoir; in hot summer, development can be even faster.

2. In the "Discussion" section of this manuscript, there are weak citations of articles that examine the impact of climate on the populations of various groups of animals, and articles with predictive models of the resistance of various animal species to climate fluctuations in the future.

Author Response

In the attached document we have tried to answer your questions.

Thank you very much.

Round 2

Reviewer 1 Report

The newly created revisions meet my requirements, but they were apparently created in a big hurry and need some fine-tuning.

L61-L63 and L64-L66 are repeated, reduce it.

L68: S. flaveolum should be in italics

I don't like the level of use of the information I gave the authors regarding the presence of species in various habitat types in Central Europe. For 1) I thought they would use this comment to reconsider the alleged association of the species with high mountains areas (not only high mountains), and the alleged stenosis of the species (L84, in fact, the species is quite eurytopic), and sensitively edit this information across the manuscript. Instead, we got a hard copy into a certain place of the manuscript. 2) If I were to accept this solution, the problem arises that the ecology of the species seems to me now inappropriately fragmented in several paragraphs: It starts to be talked about at L68, then the ecology of the species is discussed from L107, at L109-L111 there is additional information about the use of different water bodies, but on L115 the information that the nymphs are found in oligotrophic lakes remains in partial opposition. Why not merge at least L109-L111 with L115 in some more sensitive way? 3) It is advisable to support the added statement with literature - even at the price that you will not be able to find references for every piece of information I mentioned and you will have to reduce the added sentence.

Working with data from GBIF - I understand this limitation and I'm fine with it. But don't try to argue with me that "Besides, the year-based data assessment is not the purpose of the project." - The whole purpose of this project is to predict the future distribution of some species, in a resolution of twenty-year periods. Theoretically (although I understand that it is practically difficult to implement) this distinction should be used retroactively. Again, I'm ok with your solution, but this particular argument is nonsensical.

Author Response

The newly created revisions meet my requirements, but they were apparently created in a big hurry and need some fine-tuning.

Thank you very much for your recommendations, which have significatly improved the quality of the submitted paper.

L61-L63 and L64-L66 are repeated, reduce it.

Thank you for your comment. We have erased the repeated sentences.

L68: S. flaveolum should be in italics

Thank you for finding that mistake. We have corrected it in the new version.

I don't like the level of use of the information I gave the authors regarding the presence of species in various habitat types in Central Europe.

For 1) I thought they would use this comment to reconsider the alleged association of the species with high mountains areas (not only high mountains), and the alleged stenosis of the species (L84, in fact, the species is quite eurytopic), and sensitively edit this information across the manuscript. Instead, we got a hard copy into a certain place of the manuscript.

Thank you for your comment. Following the reviewers' recommendation we added this information in the last version of the manuscript. We are aware that sometimes it is had to include the recommendations from different reviewers with different opinions.

2) If I were to accept this solution, the problem arises that the ecology of the species seems to me now inappropriately fragmented in several paragraphs: It starts to be talked about at L68, then the ecology of the species is discussed from L107, at L109-L111 there is additional information about the use of different water bodies, but on L115 the information that the nymphs are found in oligotrophic lakes remains in partial opposition. Why not merge at least L109-L111 with L115 in some more sensitive way?

Thank you for your comment. We have merged L109-111 with 115 to make that paragraph more understandable. 

3) It is advisable to support the added statement with literature - even at the price that you will not be able to find references for every piece of information I mentioned and you will have to reduce the added sentence.

Thank you. We included citation for that statement (reference numbers 38 and 39).

Working with data from GBIF - I understand this limitation and I'm fine with it. But don't try to argue with me that "Besides, the year-based data assessment is not the purpose of the project." - The whole purpose of this project is to predict the future distribution of some species, in a resolution of twenty-year periods. Theoretically (although I understand that it is practically difficult to implement) this distinction should be used retroactively. Again, I'm ok with your solution, but this particular argument is nonsensical.

Thank you for your comments and for agreeing with the methods used within this paper.